# The Sample Complexity of One-Hidden-Layer Neural Networks

**Gal Vardi**
TTI Chicago and Hebrew University*
galvardi@ttic.edu

**Ohad Shamir**
Weizmann Institute of Science
ohad.shamir@weizmann.ac.il

**Nathan Srebro**
TTI Chicago
nati@ttic.edu

## Abstract

We study norm-based uniform convergence bounds for neural networks, aiming at a tight understanding of how these are affected by the architecture and type of norm constraint, for the simple class of scalar-valued one-hidden-layer networks, and inputs bounded in Euclidean norm. We begin by proving that in general, controlling the spectral norm of the hidden layer weight matrix is insufficient to get uniform convergence guarantees (independent of the network width), while a stronger Frobenius norm control is sufficient, extending and improving on previous work. Motivated by the proof constructions, we identify and analyze two important settings where (perhaps surprisingly) a mere spectral norm control turns out to be sufficient: First, when the network's activation functions are sufficiently smooth (with the result extending to deeper networks); and second, for certain types of convolutional networks. In the latter setting, we study how the sample complexity is additionally affected by parameters such as the amount of overlap between patches and the overall number of patches.

## 1 Introduction

Understanding why large neural networks are able to generalize is one of the most important puzzles in the theory of deep learning. Since sufficiently large neural networks can approximate any function, their success must be due to a strong inductive bias in the learned network weights, which is still not fully understood.

A useful approach to understand such biases is studying what types of constraints on the network weights can lead to uniform convergence bounds, which ensure that empirical risk minimization will not lead to overfitting. Notwithstanding the ongoing debate on whether uniform convergence can fully explain the learning performance of neural networks [Nagarajan and Kolter, 2019, Negrea et al., 2020, Koehler et al., 2021], these bounds provide us with important insights on what norm-based biases can potentially aid in generalization. For example, for linear predictors, it is well-understood that constraints on the Euclidean norm of the weights imply uniform convergence guarantees independent of the number of parameters. This indicates that minimizing the Euclidean norm (without worrying about the number of parameters) is often a useful inductive bias, whether used explicitly or implicitly, or whether uniform convergence formally holds or not for some specific setup. However, neural networks have a more complicated structure than linear predictors, and we still lack a good understanding of what norm-based constraints imply a good inductive bias.

---

*Work done while the author was at the Weizmann Institute of Science

36th Conference on Neural Information Processing Systems (NeurIPS 2022).

In this paper, we study this question in the simple case of scalar-valued one-hidden-layer neural networks, which generally compute functions from $\mathbb{R}^d$ to $\mathbb{R}$ of the form $\mathbf{x} \mapsto \mathbf{u}^\top \sigma(W\mathbf{x})$, with weight matrix $W \in \mathbb{R}^{n \times d}$, weight vector $\mathbf{u}$, and a fixed (generally non-linear) activation function $\sigma$. We focus on an Euclidean setting, where the inputs $\mathbf{x}$ and output weight vector $\mathbf{v}$ are assumed to have bounded Euclidean norm. Our goal is to understand what kind of norm control on the matrix $W$ is required to achieve uniform convergence guarantees, independent of the underlying distribution and the network width $n$ (i.e., the number of neurons). Previous work clearly indicates that a bound on the spectral norm is generally necessary, but (as we discuss below) does not conclusively imply whether it is also sufficient.

Our first contribution (in Subsection 3.1) is formally establishing that spectral norm control is generally insufficient to get width-independent sample complexity bounds in high dimensions, by directly lower bounding the fat-shattering number of the predictor class. On the flip side, if we assume that the *Frobenius* norm of $W$ is bounded, then we can prove uniform convergence guarantees, independent of the network width or input dimension. The latter result is based on Rademacher complexity, and extends previous results (e.g., [Neyshabur et al., 2015, Golowich et al., 2018], which crucially required homogeneous activations) to general Lipschitz activations. In Subsection 3.2, we also prove a variant of our lower bound in the case where the input dimension is fixed, pointing at a possibly interesting regime for which good upper bounds are currently lacking.

The constructions used in these lower bounds crucially require activation functions which are non-smooth around $0$, and arbitrary weight matrices $W$ with bounded norm. Motivated by this, we identify and analyze two important settings where (perhaps surprisingly) these lower bounds can be circumvented, and where a mere spectral norm control *is* sufficient to obtain width-independent guarantees:

- The first case (studied in Sec. 4) is for networks where the activation function $\sigma$ is sufficiently smooth: Specifically, when it is analytic and the coefficients of its Taylor expansion decay sufficiently rapidly. Some examples include polynomial activations, sigmoidal functions such as the error function, and appropriate smoothings of the ReLU function. Perhaps surprisingly, the mere smoothness of the activation allows us to prove uniform convergence guarantees that depend only on the spectral norm of $W$ and the structure of the activation function, independent of the network width. Moreover, we can extend our results for deeper networks when the activations is a power function (e.g., quadratic activations).
- A second important case (studied in Sec. 5) is when the network employs weight-sharing on $W$, as in convolutional networks. Specifically, we consider two variants of one-hidden-layer convolutional networks, one with a linear output layer, and another employing max-pooling. In both cases, we present bounds on the sample complexity that depend only on the spectral norm, and study how they depend on the convolutional architecture of the network (such as the number of patches or their amount of overlap).

Our work leaves open quite a few questions and possible avenues for future research, which we discuss in Sec. 6. For lack of space, all proofs of our results appear in the appendix.

**Related Work**

The literature on the sample complexity of neural networks has rapidly expanded in recent years, and cannot be reasonably surveyed here. In what follows, we discuss only works which deal specifically with the issues we focus on in this paper.

**Frobenius vs. spectral norm Control, lower bounds.** Fat-shattering lower bounds for neural networks were developed in Anthony and Bartlett [1999], but involve size or dimension dependencies rather than norm control. Bartlett et al. [2017] proved a lower bound on the Rademacher complexity of neural networks, implying that a dependence on the spectral norm is generally necessary. Golowich et al. [2018] extended this to show that a dependence on the network width is also necessary, if only the spectral norm is controlled. However, their construction requires a vector-valued (rather than scalar-valued) output. More importantly, the lower bound is on the Rademacher complexity of the predictor class rather than the fat-shattering dimension, and thus (as we further discuss below) does not necessarily imply that the actual sample complexity with some bounded loss function indeed suffers from such a width dependence. Daniely and Granot [2019] do provide a fat-shattering lower bound, which implies that neural networks on $\mathbb{R}^d$ with bounded spectral norm and width at most

$d$ can shatter $\tilde{\Omega}(d^2)$ points with constant margin, assuming that the inputs have norm at most $\sqrt{d}$. However, this lower bound does not separate between the input norm bound and the width of the hidden layer (which both scale with $d$), and thus does not clarify the contribution of the network width to the bound. Moreover, their proof technique appears to crucially rely on the input's norm scaling with the dimension, rather than being an independent parameter.

**Frobenius vs. spectral norm control, upper bounds.** A width-independent uniform convergence guarantee, depending on the Frobenius norm, has been established in Neyshabur et al. [2015] for constant-depth networks, and in Golowich et al. [2018] for arbitrary-depth networks. However, these bounds are specific to homogeneous activation functions, whereas we tackle general Lipschitz activations (at least for one-hidden layer networks). Bounds based on other norms include Anthony and Bartlett [1999], Bartlett et al. [2017], Liang [2016], but are potentially more restrictive than the Frobenius norm, or do not lead to width-independence. Also, we note that the bound of Bartlett et al. [2017] has the nice property of depending on the distance to some fixed reference matrix, rather than the norm itself. However, we do not pursue this generalization here as it is not the focus of our work.

**Sample complexity with smooth activations.** The Rademacher complexity for networks with quadratic activations has been studied in Du and Lee [2018], but assuming Frobenius norm constraints, whereas we show that mere spectral norm constraint is sufficient to bound the Rademacher complexity independent of the network width. The strong influence of the activation function on the sample complexity has been observed in the context of VC-dimension bounds for binary classification (see Anthony and Bartlett [1999, Section 7.2]). However, we are not aware of previous results showing how the smoothness of the activation functions provably affects scale-sensitive bounds such as the Rademacher complexity in our setting.

**Sample complexity of convolutional networks.** Norm-based bounds for convolutional networks (including more general ones than the one we study) have been provided in Du et al. [2018], Long and Sedghi [2019]. However, these bounds either depend on the overall number of parameters, or apply only to average-pooling. For convolutional networks with max-pooling, Ledent et al. [2021] provide a norm-based analysis which we build on (see Sec. 5 for details). Cao and Gu [2019] showed an algorithm-dependent sample complexity of learning one-hidden-layer convolutional networks with non-overlapping filters and general activation functions. Additional works studying the generalization performance of convolutional networks in settings different than ours include Li et al. [2018], Arora et al. [2018], Wei and Ma [2019], Hsu et al. [2020], Brutzkus and Globerson [2021].

## 2   Preliminaries

**Notation.** We use bold-face letters to denote vectors, and let $[m]$ be shorthand for $\{1, \ldots, m\}$. Given a matrix $M$, $M_{i,j}$ is the entry in row $i$ and column $j$. Given a function $\sigma(\cdot)$ on $\mathbb{R}$, we somewhat abuse notation and let $\sigma(\mathbf{x})$ (for a vector $\mathbf{x}$) or $\sigma(M)$ (for a matrix $M$) denote applying $\sigma$ element-wise. A special case is when $\sigma(\cdot) = [\cdot]_+ = \max\{\cdot, 0\}$ is the ReLU function. We use standard big-Oh notation, with $\Omega(\cdot), \Theta(\cdot), \mathcal{O}(\cdot)$ hiding constants and $\tilde{\Omega}(\cdot), \tilde{\Theta}(\cdot), \tilde{\mathcal{O}}(\cdot)$ hiding constants and factors polylogarithmic in the problem parameters.

**Norms.** $\|\cdot\|$ denotes the operator norm: For vectors, it is the Euclidean norm, and for matrices, the spectral norm (i.e., $\|M\| = \sup_{\mathbf{x}:\|\mathbf{x}\|=1} \|M\mathbf{x}\|$). $\|\cdot\|_F$ denotes the Frobenius norm (i.e., $\|M\|_F = \sqrt{\sum_{i,j} M_{i,j}^2}$ ). It is well-known that for any matrix $M$, $\|M\| \leq \|M\|_F$, so the class of matrices whose Frobenius norm is bounded by some $B$ is a subset of the class of matrices whose spectral norm is bounded by the same $B$. Moreover, if $M$ is an $n \times d$ matrix, then $\|M\|_F \leq \|M\| \cdot \sqrt{\min\{n, d\}}$.

**Network Architecture.** Most of our results pertain to scalar-valued one-hidden-layer networks, of the form $\mathbf{x} \mapsto \mathbf{u}^\top \sigma(W\mathbf{x})$, where $\mathbf{x} \in \mathbb{R}^d$, $W \in \mathbb{R}^{n \times d}$, $\mathbf{u}$ is a vector and $\sigma(\cdot)$ is some fixed non-linear function. The *width* of the network is $n$, the number of rows of $W$ (or equivalently, the number of neurons in the hidden layer of the network).

**Fat-Shattering and Rademacher Complexity.** When studying lower bounds on the sample complexity of a given function class, we use the following version of its *fat-shattering* dimension:

**Definition 1.** *A class of functions $\mathcal{F}$ on an input domain $\mathcal{X}$ shatters $m$ points $\{\mathbf{x}_i\}_{i=1}^m \subseteq \mathcal{X}$ with margin $\epsilon$, if there exist a number $s$, such that for all $\mathbf{y} \in \{0, 1\}^m$ we can find some $f \in \mathcal{F}$ such that*

*for all $i \in [m]$, $f(\mathbf{x}_i) \leq s - \epsilon$ if $y_i = 0$ and $f(\mathbf{x}_i) \geq s + \epsilon$ if $y_i = 1$. The* fat-shattering dimension *of $\mathcal{F}$ (at scale $\epsilon$) is the cardinality $m$ of the largest set of points in $\mathcal{X}$ for which the above holds.*

It is well-known that the fat-shattering dimension lower bounds the number of samples needed to learn in a distribution-free learning setting, up to accuracy $\epsilon$ (see for example Anthony and Bartlett [1999, Part III]). Thus, by proving the existence of a large set of points shattered by the function class, we get lower bounds on the fat-shattering dimension, and thus on the sample complexity.

As to upper bounds on the sample complexity, our results utilize the *Rademacher complexity* of a function class $\mathcal{F}$, which for our purposes can be defined as $\mathcal{R}_m(\mathcal{F}) = \sup_{\{\mathbf{x}_i\}_{i=1}^m \subseteq \mathcal{X}} \mathbb{E}_{\boldsymbol{\epsilon}} \left[ \sup_{f \in \mathcal{F}} \frac{1}{m} \sum_{i=1}^m \epsilon_i f_i(\mathbf{x}_i) \right]$, where $\boldsymbol{\epsilon} = (\epsilon_1, \ldots, \epsilon_m)$ is a vector of $m$ independent random variables $\epsilon_i$ uniformly distributed on $\{-1, +1\}$. Upper bounds on the Rademacher complexity directly translate to upper bounds on the sample complexity required for learning $\mathcal{F}$: Specifically, the number of inputs $m$ required to make $\mathcal{R}_m(\mathcal{F})$ smaller than some $\epsilon$ is generally an upper bound on the number of samples required to learn $\mathcal{F}$ up to accuracy $\epsilon$, using any Lipschitz loss (see Bartlett and Mendelson [2002], Shalev-Shwartz and Ben-David [2014], Mohri et al. [2018]). We note that Rademacher complexity bounds can also be easily converted to *margin-based* bounds (where the $0-1$ classification risk is upper-bounded by the proportion of margin violations on the training data) by considering a composition of the hypothesis class with an appropriate ramp loss (which upper bounds the 0-1 loss and lower bounds the margin loss, as was done for example in Bartlett and Mendelson [2002], Bartlett et al. [2017]).

We note that although the fat-shattering dimension and Rademacher complexity of the predictor class are closely related, they do no always behave the same: For example, the class of norm-bounded linear predictors $\{\mathbf{x} \mapsto \langle \mathbf{w}, \mathbf{x} \rangle : \mathbf{w} \in \mathbb{R}^d, \|\mathbf{w}\| \leq B\}$ has Rademacher complexity $\Theta(B/\sqrt{m})$, implying $\Theta((B/\epsilon)^2)$ samples to make it less than $\epsilon$. In contrast, the fat-shattering dimension of the class is smaller, $\Theta(\min\{d, (B/\epsilon)^2\})$ [Anthony and Bartlett, 1999, Bartlett and Mendelson, 2002]. The reason for this discrepancy is that the Rademacher complexity of the predictor class necessarily scales with the range of the function outputs, which is not necessarily relevant if we use bounded losses (that is, if we are actually interested in the function class of linear predictors composed with a bounded loss). Such bounded losses are common, for example, when we are interested in bounding the expected misclassification error (see for example Bartlett and Mendelson [2002], Bartlett et al. [2017]). For this reason, when considering the predictor class itself, we focus on fat-shattering dimension in our lower bounds, and Rademacher complexity in our upper bounds.

## 3  Frobenius Norm Control is Necessary for General Networks

We begin by considering one-hidden-layer networks $\mathbf{x} \mapsto \mathbf{u}^\top \sigma(W\mathbf{x})$, where $\sigma$ is a function on $\mathbb{R}$ applied element-wise (such as the ReLU activation function). In Subsection 3.1, we consider the dimension-free case (where we are interested in bounds that do not depend on the input dimension $d$). In Subsection 3.2, we consider the case where the dimension $d$ is a fixed parameter.

### 3.1  Dimension-Free Bounds

We focus on the following hypothesis class of scalar-valued, one-hidden-layer neural networks of width $n$ on inputs in $\mathbb{R}^d$, where $\sigma$ is a function on $\mathbb{R}$ applied element-wise, and where we only bound the operator norms:

$$\mathcal{H}^\sigma_{b,B,n,d} := \left\{ \mathbf{x} \mapsto \mathbf{u}^\top \sigma(W\mathbf{x}) \,:\, \mathbf{u} \in \mathbb{R}^n \,,\, W \in \mathbb{R}^{n \times d} \,,\, \|\mathbf{u}\| \leq b \,,\, \|W\| \leq B \right\} \,.$$

The following theorem shows that if the input dimension is large enough, then under a mild condition on the non-smoothness of $\sigma$ around 0, the fat-shattering dimension of this class necessarily scales with the network width $n$:

**Theorem 1.** *Suppose that the activation function $\sigma$ (as a function on $\mathbb{R}$) is 1-Lipschitz on $[-1, +1]$, and satisfies $\sigma(0) = 0$ as well as $\inf_{\delta \in (0,1)} \left| \frac{\sigma(\delta) + \sigma(-\delta)}{\delta} \right| \geq \alpha$ for some $\alpha > 0$.*

*Then there exist universal constants $c, c' > 0$ such that the following hold: For any $b, B, b_x, n, \epsilon > 0$, there is some $d_0 = poly(b, B, b_x, n, 1/\epsilon)$ such that for any input dimension $d \geq d_0$, $\mathcal{H}^\sigma_{b,B,n,d}$ can*

*shatter*

$$c\alpha^2 \cdot \frac{(bBb_x)^2 n}{\epsilon^2}$$

*points from* $\{\mathbf{x} \in \mathbb{R}^d : \|\mathbf{x}\| \leq b_x\}$ *with margin* $\epsilon$*, provided the expression above is larger than* $c'(\frac{1}{\alpha^2} + B^2 + n)$.

To understand the condition on $\sigma$ in the theorem, suppose that $\sigma$ has a left-hand derivative $\sigma'_-(0)$ and right-hand derivative $\sigma'_+(0)$ at 0. Recalling that $\sigma(0) = 0$, the condition stated in the theorem implies that $\left| \frac{\sigma(\delta) - \sigma(0)}{\delta} - \frac{\sigma(0) - \sigma(-\delta)}{\delta} \right| \geq \alpha$ for all $\delta > 0$. In particular, as $\delta \to 0$, we get $|\sigma'_+(0) - \sigma'_-(0)| > \alpha$. Thus, $\sigma$ is necessarily non-differentiable at 0. For example, the ReLU activation function satisfies the assumption in the theorem with $\alpha = 1$, and the leaky ReLU function $\sigma(z) = \beta z + (1 - \beta)[z]_+$ (with parameter $\beta$) satisfies the assumption with $\alpha = 1 - \beta$.

**Remark 1.** *The assumption* $\sigma(0) = 0$ *is without much loss of generality: If* $\sigma(0) \neq 0$*, then let* $\hat{\sigma}(z) := \sigma(z) - \sigma(0)$ *be a centering of* $\sigma$*, and note that our predictors can be rewritten in the form* $\mathbf{x} \mapsto \mathbf{u}^\top \hat{\sigma}(W\mathbf{x}) + \sigma(0) \cdot \mathbf{u}^\top \mathbf{1}$*. Thus, our hypothesis class is contained in the hypothesis class of predictors of the form* $\mathbf{x} \mapsto \mathbf{u}^\top \hat{\sigma}(W\mathbf{x}) + r$ *for some bounded bias parameter* $r \in \mathbb{R}$*. This bias term does not change the fat-shattering dimension, and thus is not of much interest.*

The theorem implies that with only spectral norm control (i.e. where $\|\mathbf{u}\|, \|W\|$ is bounded), it is impossible to get bounds independent of the width of the network $n$. Initially, the lower bound might appear surprising, since if the activation function $\sigma$ is the identity, $\mathcal{H}^\sigma_{b,B,n,d}$ simply contains linear predictors of norm $\leq bB$, for which the sample complexity / fat-shattering dimension is well known to be $\mathcal{O}(bB/\epsilon^2)$ in high input dimensions, completely independent of $n$ (see discussion in the previous section). Intuitively, the extra $n$ term in the lower bound comes from the fact that for random matrices $M$, $\|\sigma(M)\|$ can typically be much larger than $\|M\|$, even when $\sigma$ is a Lipschitz function satisfying $\sigma(0) = 0$. To give a concrete example, if $M$ is an $n \times n$ matrix with i.i.d. entries uniform on $\{\pm\frac{1}{\sqrt{n}}\}$, then standard concentration results imply that $\mathbb{E}[\|M\|]$ is upper-bounded by a universal constant independent of $n$, yet the matrix $\sigma(M)$ (where $\sigma$ is entry-wise absolute value) satisfies $\|\sigma(M)\| = \sqrt{n}$ (since $\sigma(M)$ is just the constant matrix with value $\frac{1}{\sqrt{n}}$ at every entry). The formal proof (in the appendix) relies on constructing a network involving random weights, so that the spectral norm is small yet the network can return sufficiently large values due to the non-linearity.

**Remark 2.** *Thm. 1 has an interesting connection to the recent work of Bubeck et al. [2021], which implies that in order to fit* $m$ *points with bounded norm using a width-*$n$ *one-hidden-layer neural network* $\mathbf{x} \mapsto \mathbf{v}^\top \sigma(W\mathbf{x})$*, the Lipschitz constant of the network (and hence* $\|\mathbf{v}\| \cdot \|W\|$*) must be generally at least* $\Omega(\sqrt{m/n})$*. The lower bound in Thm. 1 implies a related statement in the opposite direction: If we allow* $\|\mathbf{v}\| \cdot \|W\|$ *to be sufficiently larger than* $\sqrt{m/n}$*, then there exist* $m$ *points that can be shattered with constant margin. Thus, we seem to get a good characterization of the expressiveness of one-hidden layer neural networks on finite datasets, as a function of their width and the magnitude of the weights.*

Considering the lower bound, and noting that $B^2 n$ is an upper bound on $\|W\|_F^2$ which is tight in the worst-case, the bound suggests that a control over the *Frobenius norm* $\|W\|_F$ would be sufficient to get width-independent bounds. Indeed, such results were previously known when $\sigma$ is the ReLU function, or more generally, a positive-homogeneous function of degree 1 [Neyshabur et al., 2015, Golowich et al., 2018], with the proofs crucially relying on that property. In what follows, we will prove such a result for general Lipschitz functions (at least for one-hidden layer networks).

Specifically, consider the following hypothesis class, where the previous spectral norm constraint on $W$ is replaced by a Frobenius norm constraint:

$$\mathcal{F}^\sigma_{b,B,n,d} := \left\{ \mathbf{x} \mapsto \mathbf{u}^\top \sigma(W\mathbf{x}) : \mathbf{u} \in \mathbb{R}^n, W \in \mathbb{R}^{n \times d}, \|\mathbf{u}\| \leq b, \|W\|_F \leq B \right\}.$$

**Theorem 2.** *Suppose* $\sigma(\cdot)$ *(as a function on* $\mathbb{R}$*) is* $L$*-Lipschitz and* $\sigma(0) = 0$*. Then for any* $b, B, b_x, n, d, \epsilon > 0$*, the Rademacher complexity of* $\mathcal{F}^\sigma_{b,B,n,d}$ *on* $m$ *inputs from* $\{\mathbf{x} \in \mathbb{R}^d : \|\mathbf{x}\| \leq b_x\}$ *is at most* $\epsilon$*, if*

$$m \geq c \cdot \frac{(bBb_xL)^2(1 + \log^3(m))}{\epsilon^2}$$

*for some universal constant* $c > 0$*. Thus, it suffices to have* $m = \tilde{\mathcal{O}}\left( \left( \frac{bBb_xL}{\epsilon} \right)^2 \right)$*.*

The bound is indeed independent of the network width $n$. Also, the result (as an upper bound on the Rademacher complexity) is clearly tight up to log-factors, since in the special case where $\sigma(z) = L \cdot z$ and we fix $\mathbf{u} = b \cdot \mathbf{e}_1$, then $\mathcal{F}^\sigma_{b,B,n,d}$ reduces to the class of linear predictors with Euclidean norm at most $bBL$ (on data of norm at most $b_x$), whose Rademacher complexity matches the bound above up to log-factors.

**Remark 3** (Connection to Implicit Regularization). *It was recently proved that training neural networks employing homogeneous activations on losses such as the logistic loss, without any explicit regularization, gradient methods are implicitly biased towards models which minimize the squared Euclidean norm of their parameters [Lyu and Li, 2019, Ji and Telgarsky, 2020]. In our setting of one-hidden-layer networks $\mathbf{x} \mapsto \mathbf{u}^\top \sigma(W\mathbf{x})$, this reduces to $\|\mathbf{u}\|^2 + \|W\|_F^2$. For homogeneous activations, multiplying $\mathbf{u}$ by some scalar $\alpha$ and dividing $W$ by the same scalar leaves the network unchanged. Based on this observation, and the fact that $\min_{\alpha \in \mathbb{R}} \|\alpha\mathbf{u}\|^2 + \|\frac{1}{\alpha}W\|_F^2 = 2\|\mathbf{u}\| \cdot \|W\|_F$, it follows that minimizing $\|\mathbf{u}\|^2 + \|W\|_F^2$ (under any constraints on the network's outputs) is equivalent to minimizing $\|\mathbf{u}\| \cdot \|W\|_F$ (under the same constraints). Thus, gradient methods are biased towards models which minimize our bound from Thm. 2 in terms of the norms of $\mathbf{u}, W$.*

### 3.2 Dimension-Dependent Lower Bound

The bounds presented above are dimension-free, in the sense that the upper bound holds for any input dimension $d$, and the lower bound applies once $d$ is sufficiently large. However, for neural networks the case of $d$ being a fixed parameter is also of interest, since we often wish to apply large neural networks on inputs whose dimensionality is reasonably bounded (e.g., the number of pixels in an image).

For fixed $d$, and for the predictor class itself (without an additional loss composed), it is well-known that there can be a discrepancy between the fat-shattering dimension and the Rademacher complexity, even for linear predictors (see discussion in Sec. 2). Thus, although Thm. 2 is tight as a bound on the Rademacher complexity, one may conjecture that the fat-shattering dimension (and true sample complexity for bounded losses) is actually smaller for fixed $d$.

In what follows, we focus on the case of the Frobenius norm, and provide a dimension-dependent lower bound on the fat-shattering dimension. We first state the result for a ReLU activation with a bias term (Thm. 3), and then extend it to the standard ReLU activation under a slightly more stringent condition (Corollary 1).

**Theorem 3.** *For any $b, B, b_x, n, \epsilon$, and any $d$ larger than some universal constant, there exists a choice of $\beta \in [0, \tilde{\mathcal{O}}(\frac{Bb_x}{\sqrt{dn}})]$ such that the following hold: If $\sigma(z) = [z - \beta]_+$, then $\mathcal{F}^\sigma_{b,B,n,d}$ can shatter*

$$\tilde{\Omega}\left(\min\left\{nd, \frac{bBb_x}{\epsilon}\sqrt{d}\right\}\right) \tag{1}$$

*points from $\{\mathbf{x} \in \mathbb{R}^d : \|\mathbf{x}\| \le b_x\}$ with margin $\epsilon$, assuming the expression above is larger than $cd$ for some universal constant $c > 0$, and where $\tilde{\Omega}$ hides factors polylogarithmic in $d, n, b, B, b_x, \frac{1}{\epsilon}$.*

**Corollary 1.** *The lower bound of Thm. 3 also holds for the standard ReLU activation $\sigma(z) = [z]_+$, if $\beta \le \frac{Bb_x}{\sqrt{n}}$ (which happens if the input dimension $d$ is larger than a factor polylogarithmic in the problem parameters).*

The lower bound is the minimum of two terms: The first is $nd$, which is the order of the number of parameters in the network. This term is to be expected, since the fat-shattering dimension of $\mathcal{F}$ is at most the pseudodimension of $\mathcal{F}$, which indeed scales with the number of parameters $nd$ (see Anthony and Bartlett [1999], Bartlett et al. [2019]). Hence, we cannot expect to be able to shatter many more than $nd$ points. The second term is norm- and dimension-dependent, and dominates the overall lower bound if the network width $n$ is large enough. Comparing the theorem with the $\tilde{\mathcal{O}}((bBb_x/\epsilon)^2)$ upper bound from Thm. 2, it seems to suggest that having a bounded dimension $d$ may improve the sample complexity compared to the dimension-free case, with a smaller dependence on the norm bounds. However, at the moment we do not have upper bounds which match this lower bound, or even establish that bounds better than Thm. 2 are possible when the dimension $d$ is small. We leave the question of understanding the sample complexity in the fixed-dimension regime as an interesting problem for future research.

**Remark 4** (No contradiction to upper bound in Thm. 2, due to implicit bound on $d$). *Thm. 3 requires that Eq. (1) is at least order of $d$ for the lower bound to be valid. This in turn requires that $\frac{bBb_x}{\epsilon}\sqrt{d} \gg d$, or equivalently $d \ll \left(\frac{bBb_x}{\epsilon}\right)^2$. Thus, the theorem only applies when the dimension $d$ is not too large with respect to the other parameters. We note that this is to be expected: If we allow $d \gg \left(\frac{bBb_x}{\epsilon}\right)^2$ (and $n$ sufficiently large), then the lower bound in Eq. (1) will be larger than $\left(\frac{bBb_x}{\epsilon}\right)^2$, and this would violate the $\tilde{\mathcal{O}}\left((bBb_x/\epsilon)^2\right)$ upper bound implied by Thm. 2.*

## 4 Spectral Norm Control Suffices for Sufficiently Smooth Activations

The lower bounds in the previous section crucially rely on the non-smoothness of the activation functions. Thus, one may wonder whether smoothness can lead to better upper bounds. In this section, we show that perhaps surprisingly, this is indeed the case: For sufficiently smooth activations (e.g., polynomials), one can provide width-independent Rademacher complexity bounds, using only the spectral norm. Formally, we return to the class of one-hidden-layer neural networks with spectral norm constraints,

$$\mathcal{H}^\sigma_{b,B,n,d} = \left\{ \mathbf{x} \mapsto \mathbf{u}^\top \sigma(W\mathbf{x}) \; : \; \mathbf{u} \in \mathbb{R}^n \,,\; W \in \mathbb{R}^{n \times d} \,,\; \|\mathbf{u}\| \le b \,,\; \|W\| \le B \right\} \,,$$

and state the following theorem:

**Theorem 4.** *Fix some $b, B, b_x, n, d, \epsilon > 0$. Suppose $\sigma(z) = \sum_{j=1}^\infty a_j z^j$ for some $a_1, a_2, \ldots \in \mathbb{R}$, simultaneously for all $z : |z| \le Bb_x$. Then the Rademacher complexity of $\mathcal{H}^\sigma_{b,B,n,d}$ on $m$ inputs from $\{\mathbf{x} \in \mathbb{R}^d : \|\mathbf{x}\| \le b_x\}$ is at most $\epsilon$, if*

$$m \; \ge \; \left( \frac{b \cdot \tilde{\sigma}(Bb_x)}{\epsilon} \right)^2 \;\; where \;\; \tilde{\sigma}(z) := \sum_{j=1}^\infty |a_j| z^j$$

*(assuming the sum converges).*

We note that the conditions imply $\sigma(0) = 0$, which is assumed for simplicity (see Remark 1). We emphasize that this bound depends only on spectral norms of the network and properties of the activation $\sigma$. In particular, it is independent of the network width $n$ as well as the Frobenius norm of $W$. We also note that the bound is clearly tight in some cases: For example, if $\sigma(\cdot)$ is just the identity function, then $\mathcal{H}^\sigma_{b,B,n,d}$ reduces to the class of linear predictors of Euclidean norm at most $bB$, whose Rademacher complexity on inputs of norm at most $b_x$ is well-known to equal $\Theta((bBb_x/\epsilon)^2)$. This also demonstrates that the dependence on the spectral norm $B$ is necessary, even with smooth activations.

The proof of the theorem (in the appendix) depends on algebraic manipulations, which involve 'unrolling' the Rademacher complexity as a polynomial function of the network inputs, and employing a certain technical trick to simplify the resulting expression, given a bound on the spectral norm of the weight matrix.

We now turn to provide some specific examples of $\sigma(\cdot)$ and the resulting expression $\tilde{\sigma}(Bb_x)$:

**Example 1.** *If $\sigma(z)$ is a polynomial of degree $k$, then $\tilde{\sigma}(Bb_x) = \mathcal{O}((Bb_x)^k)$ for large enough $Bb_x$.*

In the example above, the output values of predictors in the class are at most $\mathcal{O}((Bb_x)^k)$, so it is not surprising that the resulting Rademacher complexity scales in the same manner.

The theorem also extends to non-polynomial activations, as long as they are sufficiently smooth (although the dependence on $Bb_x$ in $\tilde{\sigma}(Bb_x)$ generally becomes exponential). The following is an example for a sigmoidal activation based on the error function:

**Example 2.** *If $\sigma(z) = erf(rz)$ (where erf is the error function, and $r > 0$ is a scaling parameter), then $\tilde{\sigma}(Bb_x) \le \frac{2rBb_x}{\sqrt{\pi}} \exp((rBb_x)^2)$.*

See the appendix for a proof. A sigmoidal activation also allows us to define a smooth approximation of the ReLU function, to which the theorem can be applied:

**Example 3.** *If $\sigma(y) = \frac{1}{2}\left(y + \int_{z=0}^y erf(rz)dz\right)$, then $\tilde{\sigma}(Bb_x) \le \frac{Bb_x}{2} + \frac{r(Bb_x)^2}{\sqrt{\pi}} \exp((rBb_x)^2)$.*

We note that as $r \to \infty$, $\sigma(y)$ converges uniformly to the ReLU function.

Although the last two examples imply an exponential dependence on the spectral norm bound $B$ in the theorem, they still imply that for any fixed $B$, we can get a finite size-independent sample complexity (regardless of the network's width or input dimension) while controlling only the spectral norm of the weight matrices.

### 4.1 Extension to Higher Depths for Power Activations

When the activation functions are powers of the form $\sigma(z) = z^k$ for some $k$, then the previous theorem can be extended to deeper networks. To formalize this, fix integers $k \geq 1$ and $L \geq 1$, and consider a depth-$(L+1)$ network $f_{L+1}(\mathbf{x})$ (parameterized by weight matrices $W^1, W^2, \ldots, W^L$ of some arbitrary fixed dimensions, and a weight vector $\mathbf{u}$) defined recursively as follows:

$$f_0(\mathbf{x}) = \mathbf{x} \;, \;\; \forall j \in \{0, \ldots, L-1\}, \; f_{j+1}(\mathbf{x}) \;=\; (W^{j+1} f_j(\mathbf{x}))^{\circ k} \;, \;\; f_{L+1}(\mathbf{x}) = \mathbf{u}^\top f_L(\mathbf{x}) \,.$$

where $(\mathbf{v})^{\circ k}$ denotes applying the $k$-th power element-wise on a vector $\mathbf{v}$.

**Theorem 5.** *For any integers $k, L \geq 1$ and choice of matrix dimensions at each layer, consider the class of neural networks $f_{L+1}$ as above, over all weight matrices $W^1 \ldots W^L$ such that $\|W^j\| \leq B$ for all $j$, and all $\mathbf{u}$ such that $\|\mathbf{u}\| \leq b$. Then the Rademacher complexity of this class on $m$ inputs from $\{\mathbf{x} : \|\mathbf{x}\| \leq b_x\}$ is at most $\epsilon$, if*

$$m \;\geq\; \left( \frac{b \cdot B^{k + k^2 + \ldots k^L} \cdot b_x^{k^L}}{\epsilon} \right)^2 \,.$$

For constant $k$ and constant-depth networks, the sample complexity bound in the theorem is $b \cdot \text{poly}(Bb_x)/\sqrt{m}$, where $B$ bounds merely the (relatively weak) spectral norm. We also note that the exponential/doubly-exponential dependence on $k, L$ is to be expected: Even if we consider networks where each matrix is a scalar $B$, and the input is exactly $b_k$, then multiplying by $B$ and taking the $k$-th power $L-1$ times over leads to the exact same $B^{k+k^2+\ldots k^L} \cdot b_x^{k^L}$ factor. Since the Rademacher complexity depends on the scale of the outputs, such a factor is generally unavoidable. The proof of the theorem (in the appendix) builds on the proof ideas of Thm. 4, which can be extended to deeper networks at least when the activations are power functions.

## 5 Convolutional Networks

In this section, we study another important example of neural networks which circumvent our lower bounds from Sec. 3, this time by adding additional constraints on the weight matrix. Specifically, we consider one-hidden-layer *convolutional* neural networks. These networks are defined via a set of patches $\Phi = \{\phi_j\}_{j=1}^n$, where for each $j$, the patch $\phi_j : \mathbb{R}^d \mapsto \mathbb{R}^{n'}$ projects the input vector $\mathbf{x} \in \mathbb{R}^d$ into some subset of its coordinates, namely $\phi_j(\mathbf{x}) = (x_{i_1^j}, \ldots, x_{i_{n'}^j})$ for some $\{i_1^j, \ldots, i_{n'}^j\} \subseteq \{1, \ldots, d\}$. The hidden layer is parameterized by a convolutional filter vector $\mathbf{w} \in \mathbb{R}^{n'}$, and given an input $\mathbf{x}$, outputs the vector $(\sigma(\mathbf{w}^\top \phi_1(\mathbf{x})), \ldots, \sigma(\mathbf{w}^\top \phi_n(\mathbf{x}))) \in \mathbb{R}^n$, where $\sigma$ is some activation function (e.g., ReLU). Note that this can be equivalently written as $\sigma(W\mathbf{x})$, where each row $j$ of $W$ embeds the $\mathbf{w}$ vector in the coordinates corresponding to $\phi_j(\cdot)$. In what follows, we say that a matrix $W$ *conforms* to a set of patches $\Phi = \{\phi_j\}_{j=1}^n$, if there exists a vector $\mathbf{w}$ such that $(W\mathbf{x})_j = \mathbf{w}^\top \phi_j(\mathbf{x})$ for all $\mathbf{x}$. Thus, our convolutional hidden layer corresponds to a standard hidden layer (same as in previous sections), but with the additional constraint on $W$ that it must conform to a certain set of patches.

In the first subsection below, we study networks where the convolutional hidden layer is combined with a linear output layer. In the following section, we study the case where the hidden layer is combined with a fixed pooling operation. In both cases, we will get bounds that depend on the spectral norm of $W$ and the architecture of the patches.

## 5.1 Convolutional Hidden Layer + Linear Output Layer

We begin by considering convolutional networks consisting of a convolutional hidden layer (with spectral norm control and with respect to some set of patches), followed by a linear output layer:

$$\mathcal{H}_{b,B,n,d}^{\sigma,\Phi} = \{\mathbf{x} \mapsto \mathbf{u}^\top \sigma(W\mathbf{x}) \; : \; \mathbf{u} \in \mathbb{R}^n, W \in \mathbb{R}^{n\times d}, \|\mathbf{u}\| \le b \,, \|W\| \le B \,, \; W \text{ conforms to } \Phi\}$$

The following theorem shows that we can indeed obtain a Rademacher complexity bound depending only on the spectral norm of $W$, and independent of the network width $n$, under a mild assumption about the architecture of the patches:

**Theorem 6.** *Suppose $\sigma(\cdot)$ is L-Lipschitz and $\sigma(0) = 0$. Fix some set of patches $\Phi$, and let $O_\Phi$ be the maximal number of patches that any single input coordinate (in $\{1, \ldots, d\}$) appears in. Then for any $b, B, b_x, n, d, \epsilon > 0$, the Rademacher complexity of $\mathcal{H}_{b,B,n,d}^{\sigma,\Phi}$ on $m$ inputs from $\{\mathbf{x} \in \mathbb{R}^d : \|\mathbf{x}\| \le b_x\}$ is at most $\epsilon$, if*

$$m \;\ge\; 2 \cdot O_\Phi \cdot \left(\frac{bBb_xL}{\epsilon}\right)^2 \,.$$

Other than the usual parameters, the bound in the theorem also depends on the architectural parameter $O_\Phi \in \{1, \ldots, n\}$, which quantifies the amount of "overlap" between the patches. Although it can be as large as $n$ in the worst case (when some single coordinate appears in all patches), for standard convolutional architectures it is usually quite small, and does not scale with the input dimension or the total number of patches. For example, it equals $1$ if the patches are disjoint, and more generally it equals the patch size divided by the stride. Nevertheless, an interesting open question is whether the $O_\Phi$ factor in the bound can be reduced or avoided altogether.

## 5.2 Convolutional Hidden Layer + Pooling Layer

We now turn to consider a slightly different one-hidden-layer convolutional networks, where the linear output layer is replaced by a fixed pooling layer. Specifically, we consider networks of the form

$$\mathbf{x} \;\mapsto\; \rho \circ \sigma(W\mathbf{x}) \;=\; \rho\left(\sigma(\mathbf{w}^\top \phi_1(\mathbf{x})), \ldots, \sigma(\mathbf{w}^\top \phi_n(\mathbf{x}))\right),$$

where $\sigma : \mathbb{R} \to \mathbb{R}$ is an activation function as before, and $\rho : \mathbb{R}^n \to \mathbb{R}$ is 1-Lipschitz with respect to the $\ell_\infty$ *norm*. For example, $\rho(\cdot)$ may correspond to a max-pooling layer $\mathbf{z} \mapsto \max_{j \in [n]} z_j$, or to an average-pooling layer $\mathbf{z} \mapsto \frac{1}{n} \sum_{j \in [n]} z_j$. We define the following class of networks:

$$\mathcal{H}_{B,n,d}^{\sigma,\rho,\Phi} \;:=\; \left\{\mathbf{x} \mapsto \rho \circ \sigma(W\mathbf{x}) \; : \; W \in \mathbb{R}^{n\times d} \,, \; \|W\| \le B \,, \; W \text{ conforms to } \Phi\right\} \,.$$

This class is very closely related to a class of convolutional networks recently studied in Ledent et al. [2021] using an elegant covering number argument. Using their proof technique, we first provide a Rademacher complexity upper bound (Thm. 7 below), which depends merely on the spectral norm of $W$, as well as a *logarithmic* dependence on the network width $n$. Although a logarithmic dependence is relatively mild, one may wonder if we can remove it and get a fully width-independent bound, same as our previous results. Our main novel contribution in this section (Thm. 8) is to show that this is *not* the case: The fat-shattering dimension of the class necessarily has a $\log(n)$ factor, so the upper bound is tight up to factors polylogarithmic in the sample size $m$.

**Theorem 7.** *There exists a universal constant $c > 0$ such that the following holds. Suppose that $\sigma : \mathbb{R} \to \mathbb{R}$ is L-Lipschitz and $\sigma(0) = 0$, and that $\rho : \mathbb{R}^n \to \mathbb{R}$ is 1-Lipschitz w.r.t. $\ell_\infty$ and satisfies $\rho(\mathbf{0}) = 0$. Fix some set of patches $\Phi = \{\phi_j\}_{j=1}^n$. Then, for any $B, n, d, b_x, \epsilon > 0$, the Rademacher complexity of $\mathcal{H}_{B,n,d}^{\sigma,\rho,\Phi}$ on $m$ inputs from $\left\{\mathbf{x} \in \mathbb{R}^d \; : \; \|\phi_j(\mathbf{x})\| \le b_x \text{ for all } j \in [n]\right\}$ is at most $\epsilon$, if*

$$m \ge c \cdot \left(\frac{LBb_x}{\epsilon}\right)^2 \cdot \log^2(m) \log(mn) \,.$$

*Thus, it suffices to have $m = \tilde{\mathcal{O}}\left(\left(\frac{LBb_x}{\epsilon}\right)^2\right)$.*

For the lower bound, we focus for simplicity on the case where $\rho(\mathbf{z}) = \max_j z_j$ is a max-pooling layer, and where $\sigma$ is the ReLU function (which satisfies the conditions of Thm. 7 with $L = 1$). However, we emphasize that unlike the lower bound we proved in Sec. 3, the construction does not rely on the non-smoothness of $\sigma$, and in fact can easily be verified to apply (up to constants) for any $\sigma$ satisfying $\sigma(0) = 0$ and $\sigma(\epsilon) \ge c \cdot \epsilon$ (where $c > 0$ is a constant).

**Theorem 8.** *For any $B, n, b_x, \epsilon > 0$, there is $d, \Phi$ such that the following hold: The class $\mathcal{H}_{B,n,d}^{\sigma,\rho,\Phi}$, with $\sigma$ being the ReLU function and $\rho$ being the max function, can shatter*

$$\frac{1}{4} \cdot \left( \frac{B b_x}{\epsilon} \right)^2 \cdot \log(n)$$

*points from $\{\mathbf{x} \in \mathbb{R}^d : \|\mathbf{x}\| \leq b_x\}$ with margin $\epsilon$.*

*Moreover, this claim holds already where $\Phi$ corresponds to a convolutional layer with a constant stride $1$, in the following sense: If we view the input $\mathbf{x} \in \mathbb{R}^d$ as a vectorization of a tensor of order $p = \mathcal{O}(\log(n))$, then $\Phi$ corresponds to all patches of certain fixed dimensions $s_1 \times \ldots \times s_p$ in the tensor.*

## 6    Conclusions and Open Questions

In this paper, we studied sample complexity upper and lower bounds for one-hidden layer neural networks, based on bounding the norms of the weight matrices. We showed that in general, bounding the spectral norm cannot lead to size-independent guarantees, whereas bounding the Frobenius norm does. However, the constructions also pointed out where the lower bounds can be circumvented, and where a spectral norm control suffices for width-independent guarantees: First, when the activations are sufficiently smooth, and second, for certain types of convolutional networks.

Our work raises many open questions for future research. For example, how does having a fixed input dimension $d$ affect the sample complexity of neural networks? Our lower bound in Thm. 3 indicates small $d$ might reduce the sample complexity, but currently we do not have good upper bounds that actually establish that. In a different direction, we showed that spectral norm control does not lead to width-free guarantees with non-smooth activations, whereas such guarantees are possible with very smooth activations. Can we characterize what we can get for other activation? As to convolutional networks, we studied two particular architectures employing weight-sharing: One with a linear output layer, and one with a fixed Lipschitz pooling layer mapping to $\mathbb{R}$. Even for one-hidden-layer networks, this leaves open the question of characterizing the width-independent sample complexity of networks $\mathbf{x} \mapsto \mathbf{u}^\top \rho \circ \sigma(W\mathbf{x})$, where $W$ implements weight-sharing and $\rho$ is a pooling operator mapping to $\mathbb{R}^p$ with $p > 1$ (Ledent et al. [2021] provide upper bounds in this setting, but they are not size-independent and we conjecture that they can be improved). Beyond these, perhaps the most tantalizing open question is whether our results can be extended to deeper networks, and what types of bounds we might expect.

### Acknowledgements

This research is supported in part by European Research Council (ERC) grant 754705, and NSF-BSF award 1718970. We thank Roey Magen for spotting some bugs in the proof of Thm. 2 in a previous version of this paper, and the anonymous reviewers for helpful comments.

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
