# A   Proofs

## A.1   Proof of Thm. 1

We will assume without loss of generality that the condition $\inf_{\delta \in (0,1)} \left| \frac{\sigma(\delta) + \sigma(-\delta)}{\delta} \right| \geq \alpha$ stated in the theorem holds without an absolute value, namely

$$\inf_{\delta \in (0,1)} \frac{\sigma(\delta) + \sigma(-\delta)}{\delta} \geq \alpha . \tag{2}$$

To see why, note that if $\inf_{\delta \in (0,1)} \left| \frac{\sigma(\delta) + \sigma(-\delta)}{\delta} \right| \geq \alpha \geq 0$, then $\frac{\sigma(\delta) + \sigma(-\delta)}{\delta}$ can never change sign as a function of $\delta$ (otherwise it will be $0$ for some $\delta$). Hence, the condition implies that either $\frac{\sigma(\delta) + \sigma(-\delta)}{\delta} \geq \alpha$ for all $\delta \in (0,1)$, or that $-\frac{\sigma(\delta) + \sigma(-\delta)}{\delta} \geq \alpha$ for all $\delta \in (0,1)$. We simply choose to treat the first case, as the second case can be treated with a completely identical analysis, only flipping some of the signs.

Fix some sufficiently large dimension $d$ and integer $m \leq d$ to be chosen later. Choose $\mathbf{x}_1, \ldots, \mathbf{x}_m$ to be some $m$ orthogonal vectors of norm $b_x$ in $\mathbb{R}^d$. Let $X$ be the $d \times m$ matrix whose $i$-th column is $\mathbf{x}_i$. Given this input set, it is enough to show that there is some number $s$, such that for any $\mathbf{y} \in \{0,1\}^m$, we can find a predictor (namely, $\mathbf{u}, W$ depending on $\mathbf{y}$) in our class, such that $\|\mathbf{u}\| \leq b$, $\|W\| \leq B$, and

$$\forall i \, , \mathbf{u}^\top \sigma(W \mathbf{x}_i) \text{ is } \begin{cases} \leq s - \epsilon & y_i = 0 \\ \geq s + \epsilon & y_i = 1 \end{cases} . \tag{3}$$

We will do so as follows: We let

$$\mathbf{u} = \frac{b}{\sqrt{n}} \mathbf{1} \quad \text{and} \quad W = \frac{\delta}{b_x^2} V \mathrm{diag}(\mathbf{y}) X^\top ,$$

Where $\delta \in (0,1)$ is a certain scaling factor and $V$ is a $\pm 1$-valued matrix of size $n \times m$, both to be chosen later. In particular, we will assume that $V$ is approximately balanced, in the sense that for any column $i \in [n]$ of $V$, if $p_i$ is the portion of $+1$ entries in the column, then

$$\max_i \left| \frac{1}{2} - p_i \right| \leq \frac{\alpha}{8} . \tag{4}$$

For any $i \in [m]$, since $\mathbf{x}_1, \ldots, \mathbf{x}_m$ are orthogonal and of norm $b_x$, we have

$$\mathbf{u}^\top \sigma(W \mathbf{x}_i) = \mathbf{u}^\top \sigma\left( \frac{\delta}{b_x^2} V \mathrm{diag}(\mathbf{y}) X^\top \mathbf{x}_i \right) = \mathbf{u}^\top \sigma(\delta y_i \mathbf{v}_i) = \frac{b}{\sqrt{n}} \sum_{j=1}^n \sigma(\delta y_i V_{j,i})$$

where $\mathbf{v}_i$ is the $i$-th column of $V$, and $V_{j,i}$ is the entry of $V$ in the $j$-th row and $i$-th column. Then we have the following:

- If $y_i = 0$, this equals $b\sqrt{n}\sigma(0) = 0$.
- If $y_i = 1$, this equals $b\sqrt{n}\left( p_i \sigma(\delta) + (1 - p_i)\sigma(-\delta) \right)$, where $p_i \in [\frac{1}{2} - \frac{\alpha}{8}, \frac{1}{2} + \frac{\alpha}{8}]$ is the portion of entries in the $i$-th column of $V$ with value $+1$. Rewriting it and using Eq. (2), Eq. (4) and the fact that $\sigma(\cdot)$ is 1-Lipschitz on $[-1, +1]$, we get the expression

$$b\sqrt{n}\left( \frac{\sigma(\delta) + \sigma(-\delta)}{2} - \left( \frac{1}{2} - p_i \right)(\sigma(\delta) - \sigma(-\delta)) \right) \geq b\sqrt{n}\left( \frac{\delta\alpha}{2} - \frac{\alpha}{8} \cdot 2\delta \right) = \frac{b\sqrt{n}\delta\alpha}{4} .$$

Recalling Eq. (3), we get that by fixing $s = \frac{\sqrt{n}\delta\alpha}{8}$, we can shatter the dataset as long as

$$\frac{b\sqrt{n}\delta\alpha}{8} \geq \epsilon \quad \Rightarrow \quad \delta \geq \frac{8\epsilon}{\alpha b \sqrt{n}} . \tag{5}$$

Leaving this condition for a moment, we now turn to specify how $\delta, V$ is chosen, so as to satisfy the condition $\|W\| = \|\frac{\delta}{b_x^2} V \mathrm{diag}(\mathbf{y}) X^\top\| \leq B$. To that end, we let $V$ be any $\pm 1$-valued $n \times m$ matrix which satisfies Eq. (4) as well as $\|V\| \leq c(\sqrt{n} + \sqrt{m})$, where $c \geq 1$ is some universal constant.

Such a matrix necessarily exists assuming $m$ is sufficiently larger than $\frac{1}{\alpha^2}^2$. It then follows that $\|W\| \leq \frac{\delta}{b_x^2}\|V\| \cdot \|\text{diag}(\mathbf{y})\| \cdot \|X\| \leq \frac{\delta}{b_x^2} \cdot c(\sqrt{n} + \sqrt{m}) \cdot b_x = \frac{\delta}{b_x} \cdot c(\sqrt{n} + \sqrt{m})$. Therefore, by assuming

$$\delta \leq \frac{Bb_x}{c(\sqrt{n} + \sqrt{m})},$$

we ensure that $\|W\| \leq B$.

Collecting the conditions on $\delta$ (namely, that it is in $(0, 1)$, satisfies Eq. (5), as well as the displayed equation above), we get that there is an appropriate choice of $\delta$ and we can shatter our $m$ points, as long as $m$ is sufficiently larger than $1/\alpha^2$ and that

$$1 > \frac{Bb_x}{c(\sqrt{n} + \sqrt{m})} \geq \frac{8\epsilon}{\alpha b\sqrt{n}}.$$

The first inequality is satisfied if (say) we can choose $m \geq (Bb_x/c)^2$ (which we will indeed do in the sequel). As to the second inequality, it is certainly satisfied if $m \geq n$, as well as

$$\frac{Bb_x}{2c\sqrt{m}} \geq \frac{8\epsilon}{\alpha b\sqrt{n}} \implies m \leq \left(\frac{\alpha}{16c}\right)^2 \cdot \frac{(bBb_x)^2 n}{\epsilon^2}.$$

Thus, we can shatter any number $m$ of points up to this upper bound. Picking $m$ on this order (assuming it is sufficiently larger than $1/\alpha^2$, $B^2$ or $n$), assuming that the dimension $d$ is larger than $m$, and renaming the universal constants, the result follows.

## A.2 Proof of Thm. 2

To simplify notation, we rewrite $\sup_{\mathbf{u},W:\|\mathbf{u}\|\leq b,\|W\|_F\leq B}$ as simply $\sup_{\mathbf{u},W}$. Also, we let $\mathbf{w}_j$ denote the $j$-th row of the matrix $W$.

Fix some set of inputs $\mathbf{x}_1, \ldots, \mathbf{x}_m$ with norm at most $b_x$. The Rademacher complexity equals

$$\mathbb{E}_{\boldsymbol{\epsilon}} \sup_{\mathbf{u},W} \frac{1}{m} \sum_{i=1}^{m} \epsilon_i \mathbf{u}^\top \sigma(W\mathbf{x}_i) = \mathbb{E}_{\boldsymbol{\epsilon}} \sup_{\mathbf{u},W} \frac{1}{m}\mathbf{u}^\top \left(\sum_{i=1}^{m} \epsilon_i \sigma(W\mathbf{x}_i)\right)$$

$$= \frac{b}{m} \cdot \mathbb{E}_{\boldsymbol{\epsilon}} \sup_{W} \left\|\sum_{i=1}^{m} \epsilon_i \sigma(W\mathbf{x}_i)\right\| = \frac{b}{m} \cdot \mathbb{E}_{\boldsymbol{\epsilon}} \sup_{W} \sqrt{\sum_{j=1}^{n} \left(\sum_{i=1}^{m} \epsilon_i \sigma(\mathbf{w}_j^\top \mathbf{x}_i)\right)^2}.$$

Each matrix in the set $\{W \in \mathbb{R}^{d\times n} : \|W\|_F \leq B\}$ is composed of rows, whose sum of squared norms is at most $B^2$. Thus, the set can be equivalently defined as the set of $d \times n$ matrices, where each row $j$ equals $v_j\mathbf{w}_j$ for some $v_j > 0$, $\|\mathbf{w}\|_j \leq 1$, and $\|(v_1, \ldots, v_n)\|^2 = \|\mathbf{v}\|^2 \leq B^2$. Noting that each $v_j$ is positive, we can upper bound the expression in the displayed equation above as follows:

$$\frac{b}{m} \cdot \mathbb{E}_{\boldsymbol{\epsilon}} \sup_{\mathbf{v},\{\mathbf{w}_j\}} \sqrt{\sum_{j=1}^{n} \left(\sum_{i=1}^{m} \epsilon_i \sigma(v_j\mathbf{w}_j^\top \mathbf{x}_i)\right)^2}$$

$$= \frac{b}{m} \cdot \mathbb{E}_{\boldsymbol{\epsilon}} \sup_{\mathbf{v},\{\mathbf{w}_j\}} \sqrt{\sum_{j=1}^{n} v_j^2 \left(\sum_{i=1}^{m} \frac{\epsilon_i}{v_j} \sigma(v_j\mathbf{w}_j^\top \mathbf{x}_i)\right)^2}$$

$$\leq \frac{b}{m} \cdot \mathbb{E}_{\boldsymbol{\epsilon}} \sup_{\mathbf{v},\mathbf{v}',\{\mathbf{w}_j\}} \sqrt{\sum_{j=1}^{n} v'^2_j \left(\sum_{i=1}^{m} \frac{\epsilon_i}{v_j} \sigma(v_j\mathbf{w}_j^\top \mathbf{x}_i)\right)^2}, \tag{6}$$

where $\mathbf{v}' = (v'_1, \ldots, v'_n)$ satisfies $\|\mathbf{v}'\|^2 = \sum_{j=1}^{n} v'^2_j \leq B^2$ (note that $\mathbf{v}$ must also satisfy this constraint). Considering this constraint in Eq. (6), we see that for any choice of $\boldsymbol{\epsilon}, \mathbf{v}$ and $\mathbf{w}_1, \ldots, \mathbf{w}_n$, the supremum over $\mathbf{v}'$ is clearly attained by letting $v'_{j^*} = B$ for some $j^*$ for which

---

[2]This follows from the probabilistic method: If we pick the entries of $V$ uniformly at random, then both conditions will hold with some arbitrarily large probability (assuming $m$ is sufficiently larger than $1/\alpha^2$, see for example Seginer [2000]), hence the required matrix will result with some positive probability.

$\left( \sum_{i=1}^{m} \frac{\epsilon_i}{v_j} \sigma(v_j \mathbf{w}_j^\top \mathbf{x}_i) \right)^2$ is maximized, and $v'_j = 0$ for all $j \neq j*$. Plugging this observation back into Eq. (6) and writing the supremum constraints explicitly, we can upper bound the displayed equation by

$$\frac{bB}{m} \cdot \mathbb{E}_{\boldsymbol{\epsilon}} \sup_{\mathbf{v}:\min_j v_j > 0, \|\mathbf{v}\| \leq B} \sup_{\mathbf{w}_1,\dots\mathbf{w}_n:\max_j \|\mathbf{w}_j\| \leq 1} \max_j \left| \sum_{i=1}^{m} \frac{\epsilon_i}{v_j} \sigma(v_j \mathbf{w}_j^\top \mathbf{x}_i) \right|$$

$$= \frac{bB}{m} \cdot \mathbb{E}_{\boldsymbol{\epsilon}} \sup_{v \in (0,B], \mathbf{w}:\|\mathbf{w}\| \leq 1} \left| \sum_{i=1}^{m} \frac{\epsilon_i}{v} \sigma(v\mathbf{w}^\top \mathbf{x}_i) \right|$$

$$= \frac{bB}{m} \cdot \mathbb{E}_{\boldsymbol{\epsilon}} \sup_{v \in (0,B], \mathbf{w}:\|\mathbf{w}\| \leq 1} \left| \sum_{i=1}^{m} \epsilon_i \psi_v \left( \mathbf{w}^\top \mathbf{x}_i \right) \right| , \tag{7}$$

where $\psi_v(z) := \frac{\sigma(vz)}{v}$ for any $z \in \mathbb{R}$. Since $\sigma(\cdot)$ is $L$-Lipschitz, it follows that $\psi_{\mathbf{v}}(\cdot)$ is also $L$-Lipschitz regardless of $v$, since for any $z, z' \in \mathbb{R}$,

$$|\psi_v(z) - \psi_v(z')| = \frac{|\sigma(vz) - \sigma(vz')|}{v} \leq \frac{L|vz - vz'|}{v} = L|z - z'| .$$

Thus, the supremum over $v$ in Eq. (7) corresponds to a supremum over a class of $L$-Lipschitz functions which all equal 0 at the origin (since $\psi_v(0) = \frac{\sigma(0)}{v} = 0$ by assumption). As a result, we can upper bound Eq. (7) by

$$\frac{bB}{m} \cdot \mathbb{E}_{\boldsymbol{\epsilon}} \sup_{\psi \in \Psi_L, \mathbf{w}:\|\mathbf{w}\| \leq 1} \left| \sum_{i=1}^{m} \epsilon_i \psi \left( \mathbf{w}^\top \mathbf{x}_i \right) \right| ,$$

where $\Psi_L$ is the class of *all* $L$-Lipschitz functions which equal 0 at the origin.

To continue, it will be convenient to get rid of the absolute value in the displayed expression above. This can be done by noting that the expression equals

$$\frac{bB}{m} \cdot \mathbb{E}_{\boldsymbol{\epsilon}} \sup_{\psi \in \Psi_L, \mathbf{w}:\|\mathbf{w}\| \leq 1} \max \left\{ \sum_{i=1}^{m} \epsilon_i \psi \left( \mathbf{w}^\top \mathbf{x}_i \right) , -\sum_{i=1}^{m} \epsilon_i \psi \left( \mathbf{w}^\top \mathbf{x}_i \right) \right\}$$

$$\stackrel{(*)}{\leq} \frac{bB}{m} \cdot \mathbb{E}_{\boldsymbol{\epsilon}} \left[ \sup_{\psi \in \Psi_L, \mathbf{w}:\|\mathbf{w}\| \leq 1} \sum_{i=1}^{m} \epsilon_i \psi \left( \mathbf{w}^\top \mathbf{x}_i \right) + \sup_{\psi \in \Psi_L, \mathbf{w}:\|\mathbf{w}\| \leq 1} -\sum_{i=1}^{m} \epsilon_i \psi \left( \mathbf{w}^\top \mathbf{x}_i \right) \right]$$

$$\stackrel{(**)}{=} \frac{2bB}{m} \cdot \mathbb{E}_{\boldsymbol{\epsilon}} \sup_{\psi \in \Psi_L, \mathbf{w}:\|\mathbf{w}\| \leq 1} \sum_{i=1}^{m} \epsilon_i \psi \left( \mathbf{w}^\top \mathbf{x}_i \right) , \tag{8}$$

where $(*)$ follows from the fact that $\max\{a, b\} \leq a + b$ for non-negative $a, b$ and the observation that the supremum is always non-negative (it is only larger, say, than the specific choice of $\psi$ being the zero function), and $(**)$ is by symmetry of the function class $\Psi_L$ (if $\psi \in \Psi_L$, then $-\psi \in \Psi_L$ as well).

Considering Eq. (8), this is $2bB$ times the Rademacher complexity of the function class $\{\mathbf{x} \mapsto \psi(\mathbf{w}^\top \mathbf{x}) : \psi \in \Psi_L, \|\mathbf{w}\| \leq 1\}$. In other words, this class is a composition of all linear functions of norm at most 1, and all univariate $L$-Lipschitz functions crossing the origin. Fortunately, the Rademacher complexity of such composed classes was analyzed in Golowich et al. [2017] for a different purpose. In particular, noting that $\mathbf{w}^\top \mathbf{x}_i$ is bounded in $[-b_x, b_x]$, and applying Theorem 4 from that paper, we get that Eq. (8) is upper bounded by

$$2bB \cdot cL \left( \frac{b_x}{\sqrt{m}} + \log^{3/2}(m) \cdot \mathcal{R}_m(\mathcal{H}) \right) \tag{9}$$

for some universal constant $c > 0$, where $\mathcal{H} = \{\mathbf{x} \mapsto \mathbf{w}^\top \mathbf{x} : \|\mathbf{w}\| \leq 1\}$, and $\mathcal{R}_m(\mathcal{H})$ is the Rademacher complexity of $\mathcal{H}$.

To complete the proof, we need to employ a standard upper bound on $\hat{\mathcal{R}}_m(\mathcal{H})$ (see Bartlett and Mendelson [2002], Shalev-Shwartz and Ben-David [2014]), which we derive below for completeness:

$$\hat{\mathcal{R}}_m(\mathcal{H}) \;=\; \mathbb{E}_{\boldsymbol{\epsilon}} \sup_{h \in \mathcal{H}} \frac{1}{m} \sum_{i=1}^{m} \epsilon_i h(\mathbf{x}_i) \;=\; \frac{1}{m} \mathbb{E}_{\boldsymbol{\epsilon}} \sup_{\mathbf{w}:\|\mathbf{w}\| \leq 1} \sum_{i=1}^{m} \epsilon_i \mathbf{w}^\top \mathbf{x}_i$$

$$= \frac{1}{m} \mathbb{E}_{\boldsymbol{\epsilon}} \sup_{\mathbf{w}:\|\mathbf{w}\| \leq 1} \mathbf{w}^\top \left( \sum_{i=1}^{m} \epsilon_i \mathbf{x}_i \right) \;\overset{(*)}{=}\; \frac{1}{m} \mathbb{E}_{\boldsymbol{\epsilon}} \left\| \sum_{i=1}^{m} \epsilon_i \mathbf{x}_i \right\|$$

$$\overset{(**)}{\leq} \frac{1}{m} \sqrt{\mathbb{E}_{\boldsymbol{\epsilon}} \left\| \sum_{i=1}^{m} \epsilon_i \mathbf{x}_i \right\|^2} \;=\; \frac{1}{m} \sqrt{\mathbb{E}_{\boldsymbol{\epsilon}} \sum_{i,i'=1}^{m} \epsilon_i \epsilon_{i'} \mathbf{x}_i^\top \mathbf{x}_{i'}}$$

$$= \frac{1}{m} \sqrt{\sum_{i=1}^{m} \|\mathbf{x}_i\|^2} \;\leq\; \frac{1}{m} \sqrt{m b_x^2} \;=\; \frac{b_x}{\sqrt{m}} \;,$$

where $(*)$ is by the Cauchy-Schwarz inequality, and $(**)$ is by Jensen's inequality. Plugging this back into Eq. (9), we get the following upper bound:

$$2bB \cdot cL \left( \frac{b_x}{\sqrt{m}} + \log^{3/2}(m) \cdot \frac{b_x}{\sqrt{m}} \right) \;=\; 2cbBb_x L \cdot \frac{1 + \log^{3/2}(m)}{\sqrt{m}} \;.$$

Upper bounding this by $\epsilon$, solving for $m$ and simplifying a bit, the result follows.

### A.3  Proof of Thm. 3

We fix a number of inputs $m$ to be chosen later. We let $X$ be the $d \times m$ matrix whose $i$-th column is $\mathbf{x}_i$. We choose X to be any matrix such that the following conditions hold for some universal constant $c > 0$:

- Every entry of $X$ is in $\{\pm \frac{b_x}{\sqrt{d}}\}$ (hence $\forall i$, $\|\mathbf{x}_i\| = 1$)

- $\max_{i' \neq i} |\mathbf{x}_i^\top \mathbf{x}_{i'}| \leq c b_x^2 \sqrt{\frac{\log(d)}{d}}$

- $\|X\| \leq c b_x \left( 1 + \sqrt{\frac{m}{d}} \right)$.

The existence of such a matrix follows from the probabilistic method: If we simply choose each entry of $X$ independently and uniformly from $\{\pm \frac{1}{\sqrt{d}}\}$, then the first condition automatically holds; The second condition holds with high probability by a standard concentration of measure argument and a union bound; And the third condition holds with arbitrarily high constant probability (by Markov's inequality and the fact that $\mathbb{E}[\|\frac{\sqrt{d}}{b_x} \cdot X\|] \leq c(\sqrt{d} + \sqrt{m})$, see for example Seginer [2000]). Thus, by a union bound, a random matrix satisfies all of the above with some positive probability, hence such a matrix $X$ exists.

Given this input set, it is enough to show that for any $\mathbf{y} \in \{0,1\}^m$, we can find a predictor (namely, $\mathbf{u}, W$ depending on $\mathbf{y}$) in our class, such that $\|\mathbf{u}\| \leq b$, $\|W\| \leq B$, and

$$\forall i \,, \mathbf{u}^\top \sigma(W \mathbf{x}_i) \text{ is } \begin{cases} \leq 0 & y_i = 0 \\ \geq 2\epsilon & y_i = 1 \end{cases} . \tag{10}$$

We will do so as follows: Letting $a \geq 0, p \in [0,1]$ be some parameters to be chosen later, we let

$$\mathbf{u} = \frac{b}{\sqrt{n}} \mathbf{1} \quad \text{and} \quad W = \frac{1}{b_x^2} \cdot V \mathrm{diag}(\mathbf{y}) X^\top \;,$$

Where $V \in \mathbb{R}^{n \times m}$ is a random matrix with i.i.d. entries chosen as follows:

$$V_{k,i} = \begin{cases} 0 & \text{w.p. } 1 - p \\ a & \text{w.p. } \frac{p}{2} \\ -a & \text{w.p. } \frac{p}{2} \end{cases} .$$

Note that the condition $\|\mathbf{u}\| \leq b$ follows directly from the definition of $\mathbf{u}$. We will show that there is a way to choose the parameters $a, p$ such that the following holds: For any $\mathbf{y} \in \{0, 1\}^m$, with high probability over the choice of $V$, Eq. (10) holds as well as $\|W\| \leq B$. This implies that for any $\mathbf{y}$, there exists some fixed choice of $V$ (and hence $W$) such that $\|W\| \leq B$ as well as Eq. (10) holds, implying the theorem statement.

We break this argument into two lemmas:

**Lemma 1.** *There exists a universal constant $c' > 0$ such that the following holds: For any $\epsilon \geq 0$, $\delta \in (0, \exp(-1))$ and $\mathbf{y} \in \{0, 1\}^m$, if we assume*

$$\beta = c'a\sqrt{\frac{\log(d)}{d}}\log\left(\frac{m}{\delta}\right)\left(\sqrt{pm}+1\right)$$

*as well as $a \geq 4\beta$ and $bap\sqrt{n} \geq 8\epsilon$, then Eq. (10) holds with probability at least $1 - \delta - m\exp(-pn/16)$ over the choice of $V$.*

*Proof.* Let $\mathbf{w}_k$ be the $k$-th row of $W$. Fixing some $i \in [m]$, we have

$$\mathbf{u}^\top\sigma(W\mathbf{x}_i) = \mathbf{u}^\top[W\mathbf{x}_i - \beta]_+ = \frac{b}{\sqrt{n}}\sum_{k=1}^n[\mathbf{w}_k^\top\mathbf{x}_i - \beta]_+ = \frac{b}{\sqrt{n}}\sum_{k=1}^n\left[\sum_{i'=1}^m\frac{1}{b_x^2}V_{k,i'}y_{i'}\mathbf{x}_{i'}^\top\mathbf{x}_i - \beta\right]_+$$

$$= \frac{b}{\sqrt{n}}\sum_{k=1}^n\left[V_{k,i}y_i + \sum_{i'\neq i}\frac{1}{b_x^2}V_{k,i'}y_{i'}\mathbf{x}_{i'}^\top\mathbf{x}_i - \beta\right]_+. \tag{11}$$

Recalling the assumptions on $X$ and the random choice of $V$, note that $\sum_{i'\neq i}\frac{1}{b_x^2}V_{k,i'}y_{i'}\mathbf{x}_{i'}^\top\mathbf{x}_i$ is the sum of $m-1$ independent random variables, each with mean $0$, absolute value at most $|\frac{a}{b_x^2}y_{i'}\mathbf{x}_{i'\top}\mathbf{x}_i| \leq ac\sqrt{\frac{\log(d)}{d}}$, and standard deviation at most $\sqrt{p}\cdot ac\sqrt{\frac{\log(d)}{d}}$. Thus, by Bernstein's inequality, for any $\delta \in (0, \exp(-1))$, it holds with probability at least $1 - \delta$ that

$$\left|\sum_{i'\neq i}\frac{1}{b_x^2}V_{k,i'}y_{i'}\mathbf{x}_{i'}^\top\mathbf{x}_i\right| \leq c'\left(\sqrt{p}\cdot a\sqrt{\frac{\log(d)}{d}}\cdot\sqrt{(m-1)\log\left(\frac{1}{\delta}\right)} + a\sqrt{\frac{\log(d)}{d}}\cdot\log\left(\frac{1}{\delta}\right)\right)$$

$$\leq c'a\sqrt{\frac{\log(d)}{d}}\log\left(\frac{1}{\delta}\right)\left(\sqrt{pm}+1\right),$$

where $c' > 0$ is some universal constant. Applying a union bound over all $i \in [m]$, we get that with probability at least $1 - \delta$,

$$\max_{i\in[m]}\left|\sum_{i'\neq i}\frac{1}{b_x^2}V_{k,i'}y_{i'}\mathbf{x}_{i'}^\top\mathbf{x}_i\right| \leq c'a\sqrt{\frac{\log(d)}{d}}\log\left(\frac{m}{\delta}\right)\left(\sqrt{pm}+1\right).$$

Recalling that we choose $\beta$ to equal this upper bound, and plugging back into Eq. (11), we get that with probability at least $1 - \delta$,

$$\forall i \in [m], \quad \mathbf{u}^\top\sigma(W\mathbf{x}_i) \text{ is } \begin{cases} \leq \frac{b}{\sqrt{n}}\sum_{k=1}^n[V_{k,i}y_i]_+ = 0 & \text{if } y_i = 0 \\ \geq \frac{b}{\sqrt{n}}\sum_{k=1}^n[V_{k,i}y_i - 2\beta]_+ = \frac{b}{\sqrt{n}}\sum_{k=1}^n[V_{k,i} - 2\beta]_+ & \text{if } y_i = 1 \end{cases}.$$

Moreover, by the assumption $a \geq 4\beta$, we have

$$\frac{b}{\sqrt{n}}\sum_{k=1}^n[V_{k,i} - 2\beta]_+ \geq \frac{b}{\sqrt{n}}\sum_{k:V_{k,i}=a}\left[a - \frac{a}{2}\right]_+ \geq \frac{ba}{2\sqrt{n}}\sum_{k:V_{k,i}=a}1.$$

Note that $\mathbb{E}_V[\sum_{k:V_{k,i}=a}1] = \frac{pn}{2}$. Thus, by a standard multiplicative Chernoff bound and a union bound, $\sum_{k:V_{k,i}=a}1 \geq \frac{pn}{4}$ simultaneously for all $i \in [m]$, with probability at least $1 - m\exp(-pn/16)$. Combining with the above using a union bound, we get that with probability at least $1 - \delta - m\exp(-pn/16)$ over the choice of $V$,

$$\forall i \in [m], \quad \mathbf{u}^\top\sigma(W\mathbf{x}_i) \text{ is } \begin{cases} \leq 0 & \text{if } y_i = 0 \\ \geq \frac{bap\sqrt{n}}{4} & \text{if } y_i = 1 \end{cases}.$$

Since we assume $\frac{bap\sqrt{n}}{4} \geq 2\epsilon$, the result follows. $\qquad\square$

**Lemma 2.** *For any* $\mathbf{y} \in \{0,1\}^m$, *with probability at least* $\frac{1}{2}$ *over the random choice of* $V$, *the matrix* $W$ *satisfies*

$$\|W\|_F \le \frac{a\sqrt{2nmp}}{b_x} .$$

*Proof.* By definition of $W, V$ and $X$, we have

$$
\begin{aligned}
\mathbb{E}[\|W\|_F^2] &= \sum_{k=1}^{n}\sum_{i=1}^{d} \mathbb{E}[W_{k,i}^2] = \sum_{k=1}^{n}\sum_{i=1}^{d} \mathbb{E}\left[\left(\sum_{j=1}^{m}\frac{1}{b_x^2}V_{k,j}y_j X_{i,j}\right)^2\right] \\
&= \frac{1}{b_x^4} \cdot \sum_{k=1}^{n}\sum_{i=1}^{d} \mathbb{E}\left[\sum_{j,j'=1}^{m} V_{k,j}V_{k,j'}y_j y_{j'} X_{i,j}X_{i,j'}\right] \\
&= \frac{1}{b_x^4} \cdot \sum_{k=1}^{n}\sum_{i=1}^{d}\sum_{j=1}^{m} \mathbb{E}\left[V_{k,j}^2 y_j^2 X_{i,j}^2\right] \le \frac{1}{b_x^4} \cdot \frac{b_x^2}{d} \cdot \sum_{k=1}^{n}\sum_{i=1}^{d}\sum_{j=1}^{m} \mathbb{E}[V_{k,j}^2] \\
&= \frac{1}{b_x^2 d} \cdot ndm \cdot pa^2 = \frac{nmpa^2}{b_x^2} .
\end{aligned}
$$

By Markov's inequality, it follows that with probability at least $\frac{1}{2}$, $\|W\|_F^2 \le 2 \cdot \frac{nmpa^2}{b_x^2}$, from which the result follows. $\qquad\square$

Combining Lemma 1 and Lemma 2, and choosing $\delta = 1/4$, we get that with some positive probability over the choice of $V$, both the shattering condition in Eq. (10) holds, as well as $\|W\|_F \le B$, if the following combination of conditions are met (for suitable universal constant $c_1 > 0$):

$$m\exp\left(-\frac{pn}{16}\right) < \frac{1}{4} \ , \ \ a \ge c_1 a\sqrt{\frac{\log(d)}{d}}\log(4m)(\sqrt{pm}+1) \ , \ \ bap\sqrt{n} \ge 8\epsilon \ , \ \ a\sqrt{2nmp} \le Bb_x .$$

We now wish to choose the free parameters $p, a$, to ensure that all these conditions are met (hence we indeed manage to shatter the dataset), while allowing the size $m$ of the shattered set to be as large as possible. We begin by noting that the first condition is satisfied if $p > c_2 \frac{\log(m)}{n}$, and the second condition is satisfied if $d \ge c_3$ and $p \le c_4 \frac{d}{\log(d)\log^2(4m)m}$ (for suitable universal constants $c_2, c_3, c_4 > 0$). Thus, it is enough to require

$$d \ge c_3 \ , \ \ c_2\frac{\log(m)}{n} < p \le c_4\frac{d}{\log(d)\log^2(4m)m} \ , \ \ bap\sqrt{n} \ge 8\epsilon \ , \ \ a\sqrt{2nmp} \le Bb_x . \quad (12)$$

Let us pick in particular

$$p = c_4\frac{d}{\log(d)\log^2(4m)m}$$

(which is valid if it is in $[0,1]$ and if $c_2\frac{\log(m)}{n} \le c_4\frac{d}{\log(d)\log^2(4m)m}$, or equivalently $m\log(m)\log^2(4m) \le \frac{c_4 nd}{c_2\log(d)}$) and

$$a = \frac{8\epsilon}{bp\sqrt{n}} = \frac{8\epsilon\log(d)\log^2(4m)m}{c_4 bd\sqrt{n}}$$

(which automatically satisfied the third condition in Eq. (12)). Plugging into Eq. (12), the required conditions hold if we assume

$$d \ge c_3 \ , \ \ \frac{c_4 d}{\log(d)\log^2(4m)m} \le 1 \ , \ \ m\log^3(4m) \le \frac{c_5 nd}{\log(d)} \ , \ \ c_6\frac{\epsilon\sqrt{\log(d)}\log(4m)m}{b\sqrt{d}} \le Bb_x$$

for appropriate universal constants $c_5, c_6 > 0$. The first two conditions are satisfied if we require $m \ge c_7 d \ge c_8$ for suitable universal constants $c_7, c_8 > 0$. Thus, it is enough to require the set of conditions

$$m \ge c_6 d \ge c_7 \ , \ \ m\log^3(4m) \le \frac{c_5 nd}{\log(d)} \ , \ \ m\log(4m) \le \frac{bBb_x\sqrt{d}}{c_6\epsilon\sqrt{\log(d)}} .$$

All these conditions are satisfied if we assume $d \geq c_7/c_6$, pick

$$m = \tilde{\Theta}\left(\min\left\{nd, \frac{bBb_x}{\epsilon}\sqrt{d}\right\}\right) \qquad (13)$$

(with the $\tilde{\Theta}$ hiding constants and factors polylogarithmic in $d, n, b, B, b_x, \frac{1}{\epsilon}$)), and assume that the parameters are such that this expression is sufficiently larger than $d$, and that $d$ is larger than some universal constant.

It only remains to track what value of $\beta$ we have chosen (as a function of the problem parameters). Combining Lemma 1, the choice of $a, p$ from earlier, as well as Eq. (13), it follows that

$$\beta = \tilde{\Theta}\left(\frac{a}{\sqrt{d}}(1+\sqrt{pm})\right) = \tilde{\Theta}\left(\frac{\epsilon m}{bd\sqrt{dn}}(1+\sqrt{d})\right) = \tilde{\Theta}\left(\frac{\epsilon m}{bd\sqrt{n}}\right) = \tilde{\Theta}\left(\min\left\{\frac{\epsilon\sqrt{n}}{b}, \frac{Bb_x}{\sqrt{dn}}\right\}\right),$$

which is at most $\tilde{\mathcal{O}}(Bb_x/\sqrt{dn})$.

## A.4 Proof of Corollary 1

Thm. 3 implies that a certain dataset $\{\mathbf{x}_i\}_{i=1}^m$ of points in $\mathbb{R}^d$ of norm at most $b_x$ (where $m$ is the lower bound stated in the theorem) can be shattered with margin $\epsilon$, using networks in $\mathcal{F}_{b,B,n,d}^\sigma$ of the form $\mathbf{x} \mapsto \mathbf{u}^\top \sigma(W\mathbf{x})$, where $\sigma = [z - \beta]_+$ for some $\beta \in \left[0, \tilde{\mathcal{O}}(\frac{Bb_x}{\sqrt{dn}})\right]$. Our key observation is the following: Any network $\mathbf{x} \mapsto \mathbf{u}^\top \sigma(W\mathbf{x})$ can be equivalently written as $\tilde{\mathbf{x}} \mapsto \mathbf{u}^\top[\tilde{W}\tilde{\mathbf{x}}]_+$, where $\tilde{\mathbf{x}} = (\mathbf{x}, b_x)$, and $\tilde{W} = [W , -\frac{\beta}{b_x} \cdot \mathbf{1}]$ (namely, we add to $W$ another column with every entry being equal to $-\frac{\beta}{b_x}$. Note that if $\|\mathbf{x}\| \leq b_x$, then $\|\tilde{\mathbf{x}}\| \leq \sqrt{2}b_x$, and $\|\tilde{W}\| \leq \|W\| + \| -\frac{\beta}{b_x} \cdot \mathbf{1}\| \leq B + \frac{\beta}{b_x}\sqrt{n} \leq 2B$ under the corollary's conditions. Thus, if we can shatter a set of points $\{\mathbf{x}_i\}_{i=1}^m$ in the unit ball in $\mathbb{R}^d$ using networks from $\mathcal{F}_{b,B,n,d}^\sigma$, we can also shatter $\{\tilde{\mathbf{x}}_i\}_{i=1}^m$ in $\mathbb{R}^{d+1}$ (with norm $\leq \sqrt{2}b_x$) using networks from $\mathcal{F}_{b,2B,n,d+1}^{[\cdot]_+}$. Rescaling $b_x, B, d$ appropriately, we get the same shattering number lower bound for $\mathcal{F}_{b,B,n,d}^{[\cdot]_+}$ and inputs with norm $\leq b_x$ up to small numerical constants which get absorbed into the $\tilde{\Omega}(\cdot)$ notation.

## A.5 Proofs of Thm. 4 and Thm. 5

In what follows, given a vector $\mathbf{u}_i$, we let $u_{i,j}$ denote its $j$-th entry.

The proofs rely on the following two key technical lemmas:

**Lemma 3.** *Let $W$ be a matrix such that $\|W\| \leq 1$, with row vectors $\mathbf{w}_1, \mathbf{w}_2, \ldots$ Then the following holds for any set of vectors $\{\mathbf{u}_i\}$ with the same dimensionality as $\mathbf{w}_1$, and any scalars $\{z_{i,\ell}\}, \{z_i\}$ indexed by $i, \ell$:*

$$\sum_\ell \left(\sum_i (\mathbf{w}_\ell^\top \mathbf{u}_i) z_{i,\ell}\right)^2 \leq \sum_{\ell,r} \left(\sum_i u_{i,r} z_{i,\ell}\right)^2$$

*and*

$$\sum_\ell \left(\sum_i (\mathbf{w}_\ell^\top \mathbf{u}_i) z_i\right)^2 \leq \sum_r \left(\sum_i u_{i,r} z_i\right)^2,$$

*where the sum $r$ is over all all coordinates of each $\mathbf{u}_i$.*

*Proof.* Starting with the first inequality, the left hand side equals

$$\sum_\ell \left(\mathbf{w}_\ell^\top \left(\sum_i \mathbf{u}_i z_{i,\ell}\right)\right)^2 \leq \sum_{\ell,\ell'} \left(\mathbf{w}_{\ell'}^\top \left(\sum_i \mathbf{u}_i z_{i,\ell}\right)\right)^2 = \sum_\ell \left\|W^\top \left(\sum_i \mathbf{u}_i z_{i,\ell}\right)\right\|^2.$$

By Cauchy-Schwartz and the assumption $\|W\| \leq 1$, this is at most $\sum_\ell \|\sum_i \mathbf{u}_i z_{i,\ell}\|^2 = \sum_{\ell,r}(\sum_i u_{i,r} z_{i,\ell})^2$. As to the second inequality, the left hand side

equals

$$\sum_{\ell}\left(\mathbf{w}_{\ell}^{\top}\left(\sum_{i}\mathbf{u}_{i}z_{i}\right)\right)^{2} \;=\; \left\|W^{\top}\left(\sum_{i}\mathbf{u}_{i}z_{i}\right)\right\|^{2} \;\leq\; \left\|\sum_{i}\mathbf{u}_{i}z_{i}\right\|^{2} = \sum_{r}\left(\sum_{i}u_{i,r}z_{i}\right)^{2}$$

where we again used Cauchy Schwartz and the assumption $\|W\| \leq 1$. $\qquad\square$

**Lemma 4.** *Given a vector $\mathbf{u} \in \mathbb{R}^{d_{in}}$, a matrix $W \in \mathbb{R}^{d_{out} \times d_{in}}$ with row vectors $\mathbf{w}_1, \mathbf{w}_2, \ldots$ such that $\|W\| \leq B$, and a positive integer $k$, define*

$$f(\mathbf{u}) = (W\mathbf{u})^{\circ k} \, ,$$

*where $^{\circ k}$ denotes taking the $k$-th power element-wise. Then for any positive integer $r$, any vectors $\mathbf{u}_1, \mathbf{u}_2, \ldots$ in $\mathbb{R}^{d_{in}}$ and any scalars $\epsilon_1, \epsilon_2, \ldots$, it holds that*

$$\sum_{\ell_1,\ldots,\ell_r=1}^{d_{out}}\left(\sum_{i}\epsilon_i f(\mathbf{u}_i)_{\ell_1}\cdots f(\mathbf{u}_i)_{\ell_r}\right)^{2} \;\leq\; B^{2rk}\cdot \sum_{\ell_1,\ldots,\ell_{rk}=1}^{d_{in}}\left(\sum_{i}\epsilon_i u_{i,\ell_1}\cdots u_{i,\ell_{rk}}\right)^{2}.$$

*Proof.* It is enough to prove the result for $W$ such that $\|W\| = 1$ (and therefore $B = 1$): For any other $W$, apply the result on $\tilde{f}(\mathbf{u}) := (\frac{W}{\|W\|}\mathbf{u})^{\circ k} = \frac{1}{\|W\|^k}f(\mathbf{u})$, and rescale accordingly.

The left hand side equals

$$\sum_{\ell_1\ldots\ell_r=1}^{d_{out}}\left(\sum_{i}\epsilon_i(\mathbf{w}_{\ell_1}^{\top}\mathbf{u}_i)^{\circ k}\cdots(\mathbf{w}_{\ell_r}^{\top}\mathbf{u}_i)^{\circ k}\right)^{2} \tag{14}$$

Note that the term inside the square involves the product of $rk$ terms. We now simplify them one-by-one using Lemma 3: To start, we note that the above can be written as

$$\sum_{\ell_2\ldots\ell_r=1}^{d_{out}}\sum_{\ell_1=1}^{d_{out}}\left(\sum_{i}(\mathbf{w}_{\ell_1}^{\top}\mathbf{u}_i)\cdot\epsilon_i(\mathbf{w}_{\ell_1}^{\top}\mathbf{u}_i)^{\circ k-1}(\mathbf{w}_{\ell_2}^{\top}\mathbf{u}_i)^{\circ k}\cdots(\mathbf{w}_{\ell_r}^{\top}\mathbf{u}_i)^{\circ k}\right)^{2}$$

Denoting $\epsilon_i(\mathbf{w}_{\ell_1}^{\top}\mathbf{u}_i)^{\circ k-1}(\mathbf{w}_{\ell_2}^{\top}\mathbf{u}_i)^{\circ k}\cdots(\mathbf{w}_{\ell_r}^{\top}\mathbf{u}_i)^{\circ k}$ as $z_{i,\ell_1}$ and plugging the first inequality in Lemma 3, the above is at most

$$\sum_{\ell_2\ldots\ell_r=1}^{d_{out}}\sum_{\ell_1=1}^{d_{out}}\sum_{\ell_1'=1}^{d_{in}}\left(\sum_{i}u_{i,\ell_1'}\epsilon_i(\mathbf{w}_{\ell_1}^{\top}\mathbf{u}_i)^{\circ k-1}(\mathbf{w}_{\ell_2}^{\top}\mathbf{u}_i)^{\circ k}\cdots(\mathbf{w}_{\ell_r}^{\top}\mathbf{u}_i)^{\circ k}\right)^{2}$$

Again pulling out one of the product terms in front, we can rewrite this as

$$\sum_{\ell_2\ldots\ell_r=1}^{d_{out}}\sum_{\ell_1'=1}^{d_{in}}\sum_{\ell_1=1}^{d_{out}}\left(\sum_{i}(\mathbf{w}_{\ell_1}^{\top}\mathbf{u}_i)\cdot u_{i,\ell_1'}\epsilon_i(\mathbf{w}_{\ell_1}^{\top}\mathbf{u}_i)^{\circ k-2}(\mathbf{w}_{\ell_2}^{\top}\mathbf{u}_i)^{\circ k}\cdots(\mathbf{w}_{\ell_r}^{\top}\mathbf{u}_i)^{\circ k}\right)^{2}.$$

Again using the first inequality in Lemma 3, this is at most

$$\sum_{\ell_2\ldots\ell_r=1}^{d_{out}}\sum_{\ell_1',\ell_1''=1}^{d_{in}}\sum_{\ell_1=1}^{d_{out}}\left(\sum_{i}u_{i,\ell_1''}u_{i,\ell_1'}\epsilon_i(\mathbf{w}_{\ell_1}^{\top}\mathbf{u}_i)^{\circ k-2}(\mathbf{w}_{\ell_2}^{\top}\mathbf{u}_i)^{\circ k}\cdots(\mathbf{w}_{\ell_r}^{\top}\mathbf{u}_i)^{\circ k}\right)^{2}.$$

Repeating this to get rid of all but the last $(\mathbf{w}_{\ell_1}^{\top}\mathbf{u}_i)$ term, we get the upper bound

$$\sum_{\ell_2\ldots\ell_r=1}^{d_{out}}\sum_{\ell_1^1\ldots\ell_1^{k-1}=1}^{d_{in}}\sum_{\ell_1=1}^{d_{out}}\left(\sum_{i}u_{i,\ell_1^1}\cdots u_{i,\ell_1^{k-1}}\epsilon_i(\mathbf{w}_{\ell_1}^{\top}\mathbf{u}_i)(\mathbf{w}_{\ell_2}^{\top}\mathbf{u}_i)^{\circ k}\cdots(\mathbf{w}_{\ell_r}^{\top}\mathbf{u}_i)^{\circ k}\right)^{2}.$$

Again pulling the last $(\mathbf{w}_{\ell_1}^{\top}\mathbf{u}_i)$ term in front, and applying now the second inequality in Lemma 3 (as the remaining terms in the product no longer depend on $\ell_1$), we get the upper bound

$$\sum_{\ell_2\ldots\ell_r=1}^{d_{out}}\sum_{\ell_1^1\ldots\ell_1^k=1}^{d_{in}}\left(\sum_{i}u_{i,\ell_1^1}\cdots u_{i,\ell_1^k}\epsilon_i(\mathbf{w}_{\ell_2}^{\top}\mathbf{u}_i)^{\circ k}\cdots(\mathbf{w}_{\ell_r}^{\top}\mathbf{u}_i)^{\circ k}\right)^{2}.$$

Recalling that this is an upper bound on Eq. (14), and applying the same procedure now on the $(\mathbf{w}_{\ell_2}^\top \mathbf{u}_i), (\mathbf{w}_{\ell_3}^\top \mathbf{u}_i) \dots$ terms, we get overall an upper bound of the form

$$\sum_{\ell_1^1 \dots \ell_1^k = 1}^{d_{in}} \cdots \sum_{\ell_r^1 \dots \ell_r^k = 1}^{d_{in}} \left( \sum_i u_{i, \ell_1^1} \cdots u_{i, \ell_r^k} \epsilon_i \right)^2 .$$

Re-labeling the $rk$ indices as $\ell_1, \dots, \ell_{rk}$, the result follows. $\qquad \square$

### A.5.1 Proof of Thm. 4

Fixing a dataset $\mathbf{x}_1, \dots, \mathbf{x}_m$ and applying Cauchy-Schwartz, the Rademacher complexity is

$$\mathbb{E}_\epsilon \sup_{\mathbf{u}, W} \frac{1}{m} \sum_{i=1}^m \epsilon_i \mathbf{u}^\top \sigma(W \mathbf{x}_i) \leq \mathbb{E}_\epsilon \sup_W \frac{b}{m} \left\| \sum_{i=1}^m \epsilon_i \sigma(W \mathbf{x}_i) \right\| .$$

Recalling that $\sigma(z) = \sum_{j=1}^\infty a_j z^j$, by the triangle inequality, we have that the above is at most

$$\mathbb{E}_\epsilon \sup_W \frac{b}{m} \sum_{j=1}^\infty |a_j| \left\| \sum_{i=1}^m \epsilon_i (W \mathbf{x}_i)^j \right\| \leq \frac{b}{m} \sum_{j=1}^\infty |a_j| \mathbb{E}_\epsilon \sup_W \left\| \sum_{i=1}^m \epsilon_i (W \mathbf{x}_i)^j \right\|$$

where $(\cdot)^j$ is applied element-wise. Recalling that the supremum is over matrices of spectral norm at most $B$, and using Jensen's inequality, the above can be equivalently written as

$$\frac{b}{m} \sum_{j=1}^\infty |a_j| B^j \cdot \mathbb{E}_\epsilon \sup_{W : \|W\| \leq 1} \left\| \sum_{i=1}^m \epsilon_i (W \mathbf{x}_i)^j \right\| \leq \frac{b}{m} \sum_{j=1}^\infty |a_j| B^j \sqrt{\mathbb{E}_\epsilon \sup_{W : \|W\| \leq 1} \left\| \sum_{i=1}^m \epsilon_i (W \mathbf{x}_i)^j \right\|^2} .$$
(15)

Using Lemma 4, we have that for any $W : \|W\| \leq 1$,

$$\left\| \sum_{i=1}^m \epsilon_i (W \mathbf{x}_i)^j \right\|^2 = \sum_\ell \left( \sum_i \epsilon_i (W \mathbf{x}_i)_\ell^j \right)^2 \leq \sum_{\ell_1, \dots, \ell_j = 1}^d \left( \sum_{i=1}^m \epsilon_i x_{i, \ell_1} \cdots x_{i, \ell_j} \right)^2 .$$

Thus,

$$\mathbb{E}_\epsilon \sup_{W : \|W\| \leq 1} \left\| \sum_{i=1}^m \epsilon_i (W \mathbf{x}_i)^j \right\|^2 \leq \mathbb{E}_\epsilon \sum_{\ell_1, \dots, \ell_j = 1}^d \left( \sum_{i=1}^m \epsilon_i x_{i, \ell_1} \cdots x_{i, \ell_j} \right)^2$$

$$= \mathbb{E}_\epsilon \sum_{i, i' = 1}^m \sum_{\ell_1, \dots, \ell_j = 1}^d \epsilon_i \epsilon_{i'} x_{i, \ell_1} x_{i', \ell_1} \cdots x_{i, \ell_j} x_{i', \ell_j}$$

$$\stackrel{(*)}{=} \sum_{i=1}^m \sum_{\ell_1, \dots, \ell_j = 1}^d x_{i, \ell_1}^2 \cdots x_{i, \ell_j}^2$$

$$= \sum_{i=1}^m \left( \sum_{\ell_1 = 1}^d x_{i, \ell_1}^2 \right) \cdots \left( \sum_{\ell_j = 1}^d x_{i, \ell_j}^2 \right)$$

$$= \sum_{i=1}^m \|\mathbf{x}_i\|^{2j} \leq \sum_{i=1}^m b_x^{2j} = m \cdot b_x^{2j} ,$$

where in $(*)$ we used the fact that each $\epsilon_i$ is independent and uniformly distributed on $\pm 1$. Plugging this bound back into Eq. (15), we get that the Rademacher complexity is at most

$$\frac{b}{m} \sum_{j=1}^\infty |a_j| (B b_x)^j \sqrt{m} = \frac{b \cdot \tilde{\sigma}(B b_x)}{\sqrt{m}} .$$

Upper bounding this by $\epsilon$ and solving for $m$, the result follows.

## A.6 Proof of Example 2

$\sigma(z) = \mathrm{erf}(rz) = \frac{2}{\sqrt{\pi}} \int_{t=0}^{rz} \exp(-t^2) dt = \frac{2}{\sqrt{\pi}} \int_{t=0}^{rz} \sum_{j=0}^{\infty} \frac{(-t^2)^j}{j!} dt = \frac{2}{\sqrt{\pi}} \sum_{j=0}^{\infty} \frac{(-1)^j (rz)^{2j+1}}{j!(2j+1)}$, and

therefore $\tilde{\sigma}(z) = \frac{2}{\sqrt{\pi}} \sum_{j=0}^{\infty} \frac{(rz)^{2j+1}}{j!(2j+1)} \leq \frac{2rz}{\sqrt{\pi}} \sum_{j=0}^{\infty} \frac{((rz)^2)^j}{j!} = \frac{2rz}{\sqrt{\pi}} \exp\left((rz)^2\right)$.

## A.7 Proof of Example 3

By a computation similar to the previous example, $\sigma(y) = \frac{1}{2}y + \frac{1}{\sqrt{\pi}} \sum_{j=0}^{\infty} \frac{(-1)^j (r^{2j+1} y^{2j+2})}{j!(2j+1)(2j+2)}$, and

therefore $\tilde{\sigma}(z) = \frac{z}{2} + \frac{1}{\sqrt{\pi}} \sum_{j=0}^{\infty} \frac{r^{2j+1} z^{2j+2}}{j!(2j+1)(2j+2)} \leq \frac{z}{2} + \frac{rz^2}{\sqrt{\pi}} \sum_{j=0}^{\infty} \frac{((rz)^2)^j}{j!} = \frac{z}{2} + \frac{rz^2}{\sqrt{\pi}} \exp((rz)^2)$.

## A.8 Proof of Thm. 5

For simplicity, we use $\sup_{\mathbf{u},W^1,\dots,W^L}$ as short for $\sup_{\mathbf{u}:\|\mathbf{u}\|\leq b, W^1,\dots,W^L:\max_j \|W^j\|\leq B}$. The Rademacher complexity equals

$$\mathbb{E}_{\boldsymbol{\epsilon}} \sup_{\mathbf{u},W^1,\dots,W^L} \frac{1}{m} \sum_{i=1}^{m} \epsilon_i f_{L+1}(\mathbf{x}_i) = \mathbb{E}_{\boldsymbol{\epsilon}} \sup_{\mathbf{u},W^1,\dots,W^L} \frac{1}{m} \sum_{i=1}^{m} \epsilon_i \mathbf{u}^\top f_L(\mathbf{x}_i)$$

$$\leq \mathbb{E}_{\boldsymbol{\epsilon}} \sup_{\mathbf{u},W^1,\dots,W^L} \mathbf{u}^\top \left( \frac{1}{m} \sum_{i=1}^{m} \epsilon_i f_L(\mathbf{x}_i) \right) \leq \frac{b}{m} \cdot \mathbb{E}_{\boldsymbol{\epsilon}} \sup_{\mathbf{u},W^1,\dots,W^L} \left\| \sum_{i=1}^{m} \epsilon_i f_L(\mathbf{x}_i) \right\|$$

$$\leq \frac{b}{m} \sqrt{\mathbb{E}_{\boldsymbol{\epsilon}} \sup_{\mathbf{u},W^1,\dots,W^L} \left\| \sum_{i=1}^{m} \epsilon_i f_L(\mathbf{x}_i) \right\|^2} = \frac{b}{m} \sqrt{\mathbb{E}_{\boldsymbol{\epsilon}} \sup_{\mathbf{u},W^1,\dots,W^L} \sum_{\ell} \left( \sum_{i=1}^{m} \epsilon_i (f_L(\mathbf{x}_i))_\ell \right)^2}, \quad (16)$$

where we used Cauchy-Schwartz and the assumption $\|\mathbf{u}\| \leq b$, and $\ell$ ranges over the indices of $f_L(\mathbf{x}_i)$. Recalling that $f_{j+1}(\mathbf{x}) = (W^{j+1} f_j(\mathbf{x}))^{\circ k}$ and repeatedly applying Lemma 4, we have

$$\sum_{\ell} \left( \sum_{i=1}^{m} \epsilon_i (f_L(\mathbf{x}_i))_\ell \right)^2 \leq \sum_{\ell} B^{2k} \sum_{\ell_1 \dots \ell_k} \left( \sum_{i=1}^{m} \epsilon_i (f_{L-1}(\mathbf{x}_i))_{\ell_1} \cdots (f_{L-1}(\mathbf{x}_i))_{\ell_k} \right)^2$$

$$\leq B^{2k+2k^2} \sum_{\ell_1 \dots \ell_{k^2}} \left( \sum_{i=1}^{m} \epsilon_i (f_{L-2}(\mathbf{x}_i))_{\ell_1} \cdots (f_{L-2}(\mathbf{x}_i))_{\ell_k} \right)^2$$

$$\leq \cdots \leq B^{2k+2k^2+\dots 2k^L} \sum_{\ell_1 \dots \ell_{k^L}} \left( \sum_{i=1}^{m} \epsilon_i (f_0(\mathbf{x}_i))_{\ell_1} \cdots (f_0(\mathbf{x}_i))_{\ell_{k^L}} \right)^2$$

$$= B^{2k+2k^2+\dots 2k^L} \sum_{\ell_1 \dots \ell_{k^L}} \left( \sum_{i=1}^{m} \epsilon_i (f_0(\mathbf{x}_i))_{\ell_1} \cdots (f_0(\mathbf{x}_i))_{\ell_{k^L}} \right)^2$$

$$= B^{2k+2k^2+\dots 2k^L} \sum_{\ell_1 \dots \ell_{k^L}} \left( \sum_{i=1}^{m} \epsilon_i x_{i,\ell_1} \cdots x_{i,\ell_{k^L}} \right)^2$$

Therefore, recalling that $\epsilon_1 \ldots \epsilon_m$ are i.i.d. and uniform on $\{-1, +1\}$, we have

$$\mathbb{E}_{\boldsymbol{\epsilon}} \sup_{\mathbf{u}, W^0, \ldots, W^{L-1}} \sum_{\ell} \left( \sum_{i=1}^{m} \epsilon_i (f_L(\mathbf{x}_i))_\ell \right)^2 \leq B^{2k + 2k^2 + \ldots 2k^L} \mathbb{E}_{\boldsymbol{\epsilon}} \sum_{\ell_1 \ldots \ell_{k^L}} \left( \sum_{i=1}^{m} \epsilon_i x_{i, \ell_1} \cdots x_{i, \ell_{k^L}} \right)^2$$

$$= B^{2k + 2k^2 + \ldots 2k^L} \mathbb{E}_{\boldsymbol{\epsilon}} \sum_{\ell_1 \ldots \ell_{k^L}} \sum_{i, i'=1}^{m} \epsilon_i \epsilon_{i'} x_{i, \ell_1} x_{i', \ell_1} \cdots x_{i, \ell_{k^L}} x_{i', \ell_{k^L}}$$

$$= B^{2k + 2k^2 + \ldots 2k^L} \sum_{\ell_1 \ldots \ell_{k^L}} \sum_{i=1}^{m} x_{i, \ell_1}^2 \cdots x_{i, \ell_{k^L}}^2$$

$$= B^{2k + 2k^2 + \ldots 2k^L} \sum_{i=1}^{m} \left( \sum_{\ell_1} x_{i, \ell_1}^2 \right) \cdots \left( \sum_{\ell_{k^L}} x_{i, \ell_{k^L}}^2 \right) \leq B^{2k + 2k^2 + \ldots 2k^L} \cdot m \cdot b_x^{2k^L} ,$$

where in the last step we used the assumption that $\|\mathbf{x}_i\|^2 \leq b_x^2$ for all $i$. Plugging this back into Eq. (16), and solving for the number of inputs $m$ required to make the expression less than $\epsilon$, the result follows.

## A.9 Proof of Thm. 6

We will need the following lemma, based on a contraction result from Ledoux and Talagrand [1991]:

**Lemma 5.** *Let $\mathcal{T}$ be a set of vectors in $\mathbb{R}^m$ which contains the origin. If $\epsilon_1, \ldots, \epsilon_m$ are i.i.d. Rademacher random variables, and $\sigma$ is an $L$-Lipschitz function on $\mathbb{R}$ with $\sigma(0) = 0$, then*

$$\mathbb{E}_{\boldsymbol{\epsilon}} \left[ \sup_{t \in \mathcal{T}} \left( \sum_{i=1}^{m} \epsilon_i \sigma(t_i) \right)^2 \right] \leq 2L^2 \cdot \mathbb{E}_{\boldsymbol{\epsilon}} \left[ \left( \sup_{t \in \mathcal{T}} \sum_{i=1}^{m} \epsilon_i t_i \right)^2 \right] .$$

*Proof.* For any realization of $\boldsymbol{\epsilon}$, $\sup_{t \in \mathcal{T}} | \sum_{i=1}^{m} \epsilon_i \sigma(t_i)|$ equals either $\sup_{t \in \mathcal{T}} \sum_{i=1}^{m} \epsilon_i \sigma(t_i)$ or $\sup_{t \in \mathcal{T}} - \sum_{i=1}^{m} \epsilon_i \sigma(t_i)$. Thus, the left hand side in the lemma can be upper bounded as follows:

$$\mathbb{E} \left[ \left( \sup_{t \in \mathcal{T}} \left| \sum_{i=1}^{m} \epsilon_i \sigma(t_i) \right| \right)^2 \right] \leq \mathbb{E} \left[ \left( \sup_{t \in \mathcal{T}} \sum_{i=1}^{m} \epsilon_i \sigma(t_i) \right)^2 + \left( \sup_{t \in \mathcal{T}} - \sum_{i=1}^{m} \epsilon_i \sigma(t_i) \right)^2 \right] .$$

Noting that $\mathbb{E}_{\boldsymbol{\epsilon}}[(\sup_{t \in \mathcal{T}} \sum_i \epsilon_i \sigma(t_i))^2]$ equals $\mathbb{E}_{\boldsymbol{\epsilon}}[(\sup_{t \in \mathcal{T}} - \sum_i \epsilon_i \sigma(t_i))^2]$ by symmetry of the $\epsilon_i$ random variables, the expression above equals

$$2 \cdot \mathbb{E} \left[ \left( \sup_{t \in \mathcal{T}} \sum_{i=1}^{m} \epsilon_i \sigma(t_i) \right)^2 \right] \stackrel{(*)}{=} 2 \cdot \mathbb{E} \left[ \left[ \sup_{t \in \mathcal{T}} \sum_{i=1}^{m} \epsilon_i \sigma(t_i) \right]_+^2 \right] = 2L^2 \cdot \mathbb{E} \left[ \left[ \sup_{t \in \mathcal{T}} \sum_{i=1}^{m} \epsilon_i \frac{1}{L} \sigma(t_i) \right]_+^2 \right] ,$$

where $(*)$ follows from the fact that the supremum is always non-negative, since $\sigma(0) = 0$ and $\mathcal{T}$ contains the origin. We now utilize equation (4.20) in Ledoux and Talagrand [1991], which implies that $\mathbb{E}_{\boldsymbol{\epsilon}} g(\sup_{t \in \mathcal{T}} \sum_i \epsilon_i \phi(t_i)) \leq \mathbb{E}_{\boldsymbol{\epsilon}} g(\sup_{t \in \mathcal{T}} \sum_i \epsilon_i t_i)$ for any 1-Lipschitz $\phi$ satisfying $\phi(0) = 0$, and any convex increasing function $g$. Plugging into the above, and using the fact that $[z]_+^2 \leq z^2$ for all $z$, the lemma follows. $\qquad \square$

We now turn to prove the theorem. The Rademacher complexity times $m$ equals

$$\mathbb{E}_{\boldsymbol{\epsilon}} \left[ \sup_{W, \mathbf{u}} \sum_{i=1}^{m} \epsilon_i \mathbf{u}^\top \sigma(W \mathbf{x}_i) \right] ,$$

where for notational convenience we drop the conditions on $W, \mathbf{u}, \mathbf{w}$ in the supremum. Using the Cauchy-Schwartz and Jensen's inequalities, this in turn can be upper bounded as follows:

$$
\mathbb{E}_{\boldsymbol{\epsilon}}\left[\sup_{W,\mathbf{u}}\mathbf{u}^{\top}\left(\sum_{i=1}^{m}\epsilon_i\sigma(W\mathbf{x}_i)\right)\right] \leq b\cdot\mathbb{E}_{\boldsymbol{\epsilon}}\left[\sup_{W}\left\|\sum_{i=1}^{m}\epsilon_i\sigma(W\mathbf{x}_i)\right\|\right]
$$

$$
\leq b\sqrt{\mathbb{E}_{\boldsymbol{\epsilon}}\left[\sup_{W}\left\|\sum_{i=1}^{m}\epsilon_i\sigma(W\mathbf{x}_i)\right\|^{2}\right]} = b\sqrt{\mathbb{E}_{\boldsymbol{\epsilon}}\left[\sup_{W}\sum_{j=1}^{n}\left(\sum_{i=1}^{m}\epsilon_i\sigma(\mathbf{w}^{\top}\phi_j(\mathbf{x}_i))\right)^{2}\right]}
$$

$$
\leq b\sqrt{\sum_{j=1}^{n}\mathbb{E}_{\boldsymbol{\epsilon}}\left[\sup_{W}\left(\sum_{i=1}^{m}\epsilon_i\sigma(\mathbf{w}^{\top}\phi_j(\mathbf{x}_i))\right)^{2}\right]}.
$$

Recall that the supremum is over all matrices $W$ which conform to the patches, and has spectral norm at most $B$. By definition, every row of this matrix has a subset of entries, which correspond to the convolutional filter vector $\mathbf{w}$. Thus, we must have $\|\mathbf{w}\| \leq B$, since the norm $\mathbf{w}$ equals the norm of any row of $W$, and the norm of a row of $W$ is a lower bound on the spectral norm. Thus, we can upper bound the expression above by taking the supremum over *all* vectors $\mathbf{w}$ such that $\|\mathbf{w}\| \leq B$ (and not just those that the corresponding matrix has spectral norm $\leq B$). Thus, we get the upper bound

$$
b\sqrt{\sum_{j=1}^{n}\mathbb{E}_{\boldsymbol{\epsilon}}\left[\sup_{\mathbf{w}:\|\mathbf{w}\|\leq B}\left(\sum_{i=1}^{m}\epsilon_i\sigma(\mathbf{w}^{\top}\phi_j(\mathbf{x}_i))\right)^{2}\right]},
$$

which by Lemma 5 and Cauchy-Shwartz, is at most

$$
bL\sqrt{2\sum_{j=1}^{n}\mathbb{E}_{\boldsymbol{\epsilon}}\left[\sup_{\mathbf{w}:\|\mathbf{w}\|\leq B}\left(\sum_{i=1}^{m}\epsilon_i\mathbf{w}^{\top}\phi_j(\mathbf{x}_i)\right)^{2}\right]} \leq bBL\sqrt{2\sum_{j=1}^{n}\mathbb{E}_{\boldsymbol{\epsilon}}\left[\left\|\sum_{i=1}^{m}\epsilon_i\phi_j(\mathbf{x}_i)\right\|^{2}\right]}
$$

$$
= bBL\sqrt{2\sum_{j=1}^{n}\mathbb{E}_{\boldsymbol{\epsilon}}\left[\sum_{i,i'=1}^{m}\epsilon_i\epsilon'_i\phi_j(\mathbf{x}_i)^{\top}\phi_j(\mathbf{x}_{i'})\right]} = bBL\sqrt{2\sum_{j=1}^{n}\sum_{i=1}^{m}\|\phi_j(\mathbf{x}_i)\|^{2}}.
$$

Recalling that $O_{\Phi}$ is the maximal number of times any single input coordinate appears across the patches, and letting $x_{i,l}$ be the $l$-th coordinate of $\mathbf{x}_i$, we can upper bound the above by

$$
bBL\sqrt{2\sum_{i=1}^{m}\sum_{l=1}^{d}x_{i,l}^{2}O_{\Phi}} = bBL\sqrt{2\sum_{i=1}^{m}\|\mathbf{x}_i\|^{2}\cdot O_{\Phi}} \leq bBb_xL\sqrt{2mO_{\Phi}}.
$$

Dividing by $m$, and solving for the number $m$ required to make the resulting expression less than $\epsilon$, the result follows.

### A.10 Proof of Thm. 7

The proof follows from a covering number argument. We start with some required definitions and lemmas.

**Definition 2.** *Let $\mathcal{F}$ be a class of functions from $\mathcal{X}$ to $\mathbb{R}$. For $1 \leq p \leq \infty$, $\epsilon > 0$, and $\{\mathbf{x}_1,\ldots,\mathbf{x}_m\} \subseteq \mathcal{X}$, the* empirical covering number $\mathcal{N}_p(\mathcal{F}, \epsilon; \mathbf{x}_1,\ldots,\mathbf{x}_m)$ *is the minimal cardinality of a set $V \subseteq \mathbb{R}^m$, such that for all $f \in \mathcal{F}$ there is $\mathbf{v} \in V$ such that*

$$
\left(\frac{1}{m}\sum_{i=1}^{m}|f(\mathbf{x}_i)-v_i|^{p}\right)^{1/p} \leq \epsilon.
$$

*We define the* covering number $\mathcal{N}_p(\mathcal{F}, \epsilon, m) = \sup_{\mathbf{x}_1,\ldots,\mathbf{x}_m}\mathcal{N}_p(\mathcal{F}, \epsilon; \mathbf{x}_1,\ldots,\mathbf{x}_m).$

**Lemma 6** (Zhang [2002]). *Let $a, b > 0$, let $\mathcal{X} = \{\mathbf{x} \in \mathbb{R}^d : \|\mathbf{x}\| \le b\}$, and consider the class of linear predictors $\mathcal{F} = \{f \in \mathbb{R}^{\mathcal{X}} : f(\mathbf{x}) = \mathbf{w}^\top \mathbf{x}, \|\mathbf{w}\| \le a\}$. Then,*

$$\log \mathcal{N}_\infty(\mathcal{F}, \epsilon, m) \le \frac{36 a^2 b^2}{\epsilon^2} \log \left(2m\lceil 4ab/\epsilon + 2\rceil + 1\right) .$$

**Lemma 7** (E.g., Daniely and Granot [2019]). *Let $C > 0$ and let $\mathcal{F}$ be a class of $C$-bounded functions from $\mathcal{X}$ to $\mathbb{R}$, i.e., $|f(\mathbf{x})| \le C$ for all $f \in \mathcal{F}$ and $\mathbf{x} \in \mathcal{X}$. Then, for every integer $M \ge 1$ we have*

$$\mathcal{R}_m(\mathcal{F}) \le C 2^{-M} + \frac{6C}{\sqrt{m}} \sum_{k=1}^{M} 2^{-k} \sqrt{\log \mathcal{N}_2(\mathcal{F}, C2^{-k}, m)} .$$

We are now ready to prove the theorem. For $i \in [m]$, $j \in [n]$ we denote $\mathbf{x}'_{i,j} = \phi_j(\mathbf{x}_i) \in \mathbb{R}^{n'}$. Let $\mathcal{X}_{n'} = \{\mathbf{x}' \in \mathbb{R}^{n'} : \|\mathbf{x}'\| \le b_x\}$, and let

$$\mathcal{F} := \{f \in \mathbb{R}^{\mathcal{X}_{n'}} : f(\mathbf{x}') = \mathbf{w}^\top \mathbf{x}', \mathbf{w} \in \mathbb{R}^{n'}, \|\mathbf{w}\| \le B\} .$$

Let $V \subseteq \mathbb{R}^{mn}$ be a set of size at most $\mathcal{N}_\infty(\mathcal{F}, \epsilon/L, mn)$, such that for all $f \in \mathcal{F}$ there is $\mathbf{v} \in V$ that satisfies the following: Letting $v_{i,j} := v_{(i-1)n+j}$, we have $|f(\mathbf{x}'_{i,j}) - v_{i,j}| \le \epsilon/L$ for all $i \in [m]$, $j \in [n]$.

We define

$$U := \{\mathbf{u} \in \mathbb{R}^m : \text{there is } \mathbf{v} \in V \text{ s.t. } u_i = \rho \circ \sigma(v_{i,1}, \dots, v_{i,n}) = \rho\left(\sigma(v_{i,1}), \dots, \sigma(v_{i,n})\right) \text{ for all } i \in [m]\} .$$

Note that $|U| \le |V|$. Let $h \in \mathcal{H}_{B,n,d}^{\sigma,\rho,\Phi}$ and suppose that the network $h$ has a filter $\mathbf{w} \in \mathbb{R}^{n'}$. Let $W$ be the weight matrix that corresponds to $\Phi$ and $\mathbf{w}$. Thus, we have $\|W\| \le B$. Let $\mathbf{x} \in \mathbb{R}^d$ such that $\phi_1(\mathbf{x}) = \frac{\mathbf{w}}{\|\mathbf{w}\|}$ and $x_k = 0$ for every coordinate $k$ that does not appear in $\phi_1$. That is, $\mathbf{x}$ is a vector of norm 1 such that $(W\mathbf{x})_1 = \mathbf{w}^\top \phi_1(\mathbf{x}) = \|\mathbf{w}\|$. Therefore, $\|W\mathbf{x}\| \ge (W\mathbf{x})_1 = \|\mathbf{w}\|$, and thus $B \ge \|W\| \ge \|\mathbf{w}\|$. Let $f$ be the function in $\mathcal{F}$ that corresponds to $\mathbf{w}$, and let $\mathbf{v} \in V$ such that $|f(\mathbf{x}'_{i,j}) - v_{i,j}| \le \epsilon/L$ for all $i \in [m]$, $j \in [n]$. Let $\mathbf{u} \in U$ that corresponds to $\mathbf{v}$, namely, $u_i = \rho \circ \sigma(v_{i,1}, \dots, v_{i,n})$ for all $i \in [m]$. Note that $|h(\mathbf{x}_i) - u_i| \le \epsilon$ for all $i \in [m]$. Indeed, we have that $|h(\mathbf{x}_i) - u_i|$ equals

$$\left|\rho \circ \sigma\left(f(\mathbf{x}'_{i,1}), \dots, f(\mathbf{x}'_{i,n})\right) - \rho \circ \sigma(v_{i,1}, \dots, v_{i,n})\right| \le L \cdot \max_{j \in [n]} \left|f(\mathbf{x}'_{i,j}) - v_{i,j}\right| \le L \cdot \frac{\epsilon}{L} = \epsilon ,$$

where the first inequality follows from the $L$-Lipschitzness of $\rho \circ \sigma$ w.r.t. $\ell_\infty$. Hence,

$$\mathcal{N}_\infty\left(\mathcal{H}_{B,n,d}^{\sigma,\rho,\Phi}, \epsilon, m\right) \le |U| \le |V| \le \mathcal{N}_\infty(\mathcal{F}, \epsilon/L, mn) .$$

Combining the above with Lemma 6, and using the fact that the $\mathcal{N}_2$ covering number is at most the $\mathcal{N}_\infty$ covering number (cf. Anthony and Bartlett [1999]), we get

$$
\begin{aligned}
\log \mathcal{N}_2\left(\mathcal{H}_{B,n,d}^{\sigma,\rho,\Phi}, \epsilon, m\right) &\le \log \mathcal{N}_\infty\left(\mathcal{H}_{B,n,d}^{\sigma,\rho,\Phi}, \epsilon, m\right) \\
&\le \log \mathcal{N}_\infty(\mathcal{F}, \epsilon/L, mn) \\
&\le \frac{36 b_x^2 B^2}{(\epsilon/L)^2} \log \left(2mn\lceil 4b_x B/(\epsilon/L) + 2\rceil + 1\right) .
\end{aligned}
\tag{17}
$$

Note that for every $\mathbf{x} \in \mathcal{X} := \{\mathbf{x} \in \mathbb{R}^d : \|\phi_j(\mathbf{x})\| \le b_x \text{ for all } j \in [n]\}$ and $h \in \mathcal{H}_{B,n,d}^{\sigma,\rho,\Phi}$ we have $|h(\mathbf{x})| = |\rho(\sigma(\mathbf{w}^\top \phi_1(\mathbf{x})), \dots, \sigma(\mathbf{w}^\top \phi_n(\mathbf{x})))| \le L b_x B$, since $|\mathbf{w}^\top \phi_j(\mathbf{x})| \le B b_x$, the activation $\sigma$ is $L$-Lipschitz and satisfies $\sigma(0) = 0$, and $\rho$ is 1-Lipschitz w.r.t. $\ell_\infty$ and satisfies $\rho(\mathbf{0}) = 0$. By Lemma 7, we conclude that

$$\mathcal{R}_m\left(\mathcal{H}_{B,n,d}^{\sigma,\rho,\Phi}\right) \le L b_x B 2^{-M} + \frac{6 L b_x B}{\sqrt{m}} \sum_{\ell=1}^{M} 2^{-\ell} \sqrt{\log \mathcal{N}_2\left(\mathcal{H}_{B,n,d}^{\sigma,\rho,\Phi}, L b_x B 2^{-\ell}, m\right)} ,$$

for every integer $M \geq 1$. By plugging-in $M = \lceil \log(\sqrt{m}) \rceil$ and the expression from Eq. (17), we get

$$\mathcal{R}_m\left(\mathcal{H}_{B,n,d}^{\sigma,\rho,\Phi}\right) \leq \frac{Lb_xB}{\sqrt{m}} + \frac{6Lb_xB}{\sqrt{m}} \sum_{\ell=1}^{\lceil\log(\sqrt{m})\rceil} 2^{-\ell} \sqrt{\frac{36b_x^2B^2}{(b_xB2^{-\ell})^2} \log\left(2mn\lceil 4b_xB/(b_xB2^{-\ell}) + 2\rceil + 1\right)}$$

$$= \frac{Lb_xB}{\sqrt{m}} + \frac{36Lb_xB}{\sqrt{m}} \sum_{\ell=1}^{\lceil\log(\sqrt{m})\rceil} \sqrt{\log\left(2mn\lceil 4 \cdot 2^\ell + 2\rceil + 1\right)}$$

$$\leq \frac{Lb_xB}{\sqrt{m}} + \frac{36Lb_xB}{\sqrt{m}} \lceil\log(\sqrt{m})\rceil \cdot \sqrt{\log\left(23mn\sqrt{m}\right)} .$$

Hence, for some universal constant $c' > 0$ the above is at most

$$c' \cdot \frac{Lb_xB\log(m)\sqrt{\log(mn)}}{\sqrt{m}} .$$

Requiring this to be at most $\epsilon$ and rearranging, the result follows.

## A.11   Proof of Thm. 8

To help the reader track the main proof ideas, we first prove the claim for the case where $B = b_x = 1$ and $\epsilon = 1/2$ (in Subsection A.11.1), and then extend the proof for arbitrary $B, b_x, \epsilon > 0$ in Subsection A.11.2.

### A.11.1   Proof for $B = b_x = 1$ and $\epsilon = 1/2$

Let $m = \log(n)$ and let $d = 3^m$. Consider $m$ points $\mathbf{x}^1, \ldots, \mathbf{x}^m$, where for every $i \in [m]$ the point $\mathbf{x}^i \in \mathbb{R}^d$ is a vectorization of an order-$m$ tensor $\hat{\mathbf{x}}^i$ such that each component is indexed by $(j_1, \ldots, j_m) \in [3]^m$. We define the components $x_{j_1,\ldots,j_m}^i$ of $\hat{\mathbf{x}}^i$ such that $x_{j_1,\ldots,j_m}^i = 1$ if $j_i = 3$, and $j_r = 2$ for all $r \neq i$, and $x_{j_1,\ldots,j_m}^i = 0$ otherwise. Note that $\|\mathbf{x}^i\| = 1$ for all $i \in [m]$. Consider patches of dimensions $2 \times \ldots \times 2$ and stride 1. Thus, the set $\Phi$ corresponds to all the patches of dimensions $2 \times \ldots \times 2$ in the tensor. Note that there are $2^m = n$ such patches. Indeed, given an index $(j_1, \ldots, j_m) \in [2]^m$, we can define a patch which contains the indices $\{(j_1, \ldots, j_m) + (\Delta_1, \ldots, \Delta_m) : (\Delta_1, \ldots, \Delta_m) \in \{0,1\}^m\}$. We say that $(j_1, \ldots, j_m)$ is the *base index* of this patch. Note that each $(j_1, \ldots, j_m) \in [2]^m$ is a base index of exactly one patch. Also, an index $(j_1, \ldots, j_m)$ which includes some $r \in [m]$ with $j_r = 3$ does not induce a patch of the form $\{(j_1, \ldots, j_m) + (\Delta_1, \ldots, \Delta_m) : (\Delta_1, \ldots, \Delta_m) \in \{0,1\}^m\}$, since for $\Delta_r = 1$ we get an invalid index.

We show that for any $\mathbf{y} \in \{0,1\}^m$ we can find a filter $\mathbf{w}$, such that $\mathbf{w}$ is an order-$m$ tensor of dimensions $2 \times \ldots \times 2$ and satisfies the following. Let $N_{\mathbf{w}}$ be the neural network that consists of a convolutional layer with the patches $\Phi$ and the filter $\mathbf{w}$, followed by a max-pooling layer. Then, $N_{\mathbf{w}}(\mathbf{x}^i) = y_i$ for all $i \in [m]$. Thus, we can shatter $\mathbf{x}^1, \ldots, \mathbf{x}^m$ with margin $\epsilon = 1/2$. Moreover, the spectral norm of the matrix $W$ that corresponds to the convolutional layer is at most 1.

Consider the filter $\mathbf{w}$ of dimensions $2 \times \ldots \times 2$ such that $w_{j_1,\ldots,j_m} = 1$ if $(j_1, \ldots, j_m) = \mathbf{1} + \mathbf{y}$, and $w_{j_1,\ldots,j_m} = 0$ otherwise. We now show that $N_{\mathbf{w}}(\mathbf{x}^i) = y_i$ for all $i \in [m]$. Since the filter $\mathbf{w}$ has a single non-zero component, then the inner product between $\mathbf{w}$ and a patch of $\mathbf{x}^i$ is non-zero iff the patch of $\mathbf{x}^i$ has a non-zero component in the appropriate position. More precisely, for a patch with base index $(j_1, \ldots, j_m)$, the inner product between the components of $\mathbf{x}^i$ in the indices of the patch and the filter $\mathbf{w}$ is 1 iff $x_{(j_1,\ldots,j_m)+\mathbf{y}}^i = 1$, and otherwise the inner product is 0. Since $x_{q_1,\ldots,q_m}^i = 1$ iff $q_i = 3$ and $q_r = 2$ for $r \neq i$, then $x_{(j_1,\ldots,j_m)+\mathbf{y}}^i = 1$ iff $j_i = 3 - y_i$ and $j_r = 2 - y_r$ for $r \neq i$. Now, if $y_i = 0$ then there is no patch such that the base index satisfies $j_i = 3 - y_i = 3$, since all base indices are in $[2]^m$, and therefore $N_{\mathbf{w}}(\mathbf{x}^i) = 0$. If $y_i = 1$ then the patch whose base index satisfies $j_i = 3 - y_i$ and $j_r = 2 - y_r$ for $r \neq i$ gives output 1 (and all other patches give output 0) and hence $N_{\mathbf{w}}(\mathbf{x}^i) = 1$. Thus, we have $N_{\mathbf{w}}(\mathbf{x}^i) = y_i$ as required.

For example, consider the case where $m = 2$. Then, the tensor $\hat{\mathbf{x}}^1$ is the matrix

$$\hat{\mathbf{x}}^1 = \begin{bmatrix} 0 & 0 & 0 \\ 0 & 0 & 0 \\ 0 & 1 & 0 \end{bmatrix} .$$

For $\mathbf{y} = (1,1)^\top$ we have $\mathbf{w} = \begin{bmatrix} 0 & 0 \\ 0 & 1 \end{bmatrix}$ and hence the patch with base index $(2,1)$ gives output 1.

For $\mathbf{y} = (1,0)^\top$ we have $\mathbf{w} = \begin{bmatrix} 0 & 0 \\ 1 & 0 \end{bmatrix}$ and hence the patch with base index $(2,2)$ gives output 1.

However, for $\mathbf{y} = (0,1)^\top$ we have $\mathbf{w} = \begin{bmatrix} 0 & 1 \\ 0 & 0 \end{bmatrix}$ and hence there is no patch that gives output 1.
Thus, in all the above cases we have $N_{\mathbf{w}}(\mathbf{x}^1) = y_1$.

It remains to show that the spectral norm of the matrix $W$ that corresponds to the convolutional layer with the filter $\mathbf{w}$ is at most 1. Thus, we show that for every input $\mathbf{x} \in \mathbb{R}^d$ with $\|\mathbf{x}\| = 1$ the inputs to the hidden layer is a vector with norm at most 1. We view $\mathbf{x}$ as the vectorization of a tensor $\hat{\mathbf{x}}$ with components $x_{j_1,\ldots,j_m}$ for $(j_1,\ldots,j_m) \in [3]^m$. Since the filter $\mathbf{w}$ contains a single 1-component and all other components are 0, then the input to each hidden neuron is a different component of $\hat{\mathbf{x}}$. More precisely, since the filter $\mathbf{w}$ contains 1 at index $\mathbf{1} + \mathbf{y}$ then for the patch with base index $(j_1,\ldots,j_m)$ the corresponding hidden neuron has input $x_{(j_1,\ldots,j_m)+\mathbf{y}}$. Note that each hidden neuron corresponds to a different base index and hence the input to each hidden neuron is a different component of $\hat{\mathbf{x}}$. Therefore, the norm of the vector whose components are the inputs to the hidden neurons is at most the norm of the input $\mathbf{x}$, and hence it is at most 1.

### A.11.2  Proof for arbitrary $B, b_x, \epsilon > 0$

Let $m = \left(\frac{b_x B}{2\epsilon}\right)^2 \cdot \log(n)$ and let $d = \left(\frac{b_x B}{2\epsilon}\right)^2 \cdot 3^{\log(n)}$. Let $m' = \log(n)$ and let $L = \left(\frac{b_x B}{2\epsilon}\right)^2$.
Consider $m$ points $\mathbf{x}^1,\ldots,\mathbf{x}^m$, where for every $i \in [m]$ the point $\mathbf{x}^i \in \mathbb{R}^d$ is a vectorization of a tensor $\hat{\mathbf{x}}^i$ of order $m'+1$, such that each component is indexed by $(j_1,\ldots,j_{m'},\ell) \in [3]^{m'} \times [L]$. Consider a partition of $[m]$ into $L$ disjoint susets $S_1,\ldots,S_L$, each of size $m/L = m'$.

We define the components $x^i_{j_1,\ldots,j_{m'},\ell}$ of $\hat{\mathbf{x}}^i$ as follows: Suppose that $i \in S_r := \{k_1,\ldots,k_{m'}\}$ for some $r \in L$, and that $i = k_t$, i.e., $i$ is the $t$-th element in the subset $S_r$. For every $\ell \neq r$ we define $x^i_{j_1,\ldots,j_{m'},\ell} = 0$ for every $j_1,\ldots,j_{m'} \in [3]^{m'}$, namely, if $\ell$ does not correspond to the subset of $i$ then the component is 0. For $\ell = r$ the component $x^i_{j_1,\ldots,j_{m'},\ell}$ is defined in a similar way to the tensor $\hat{\mathbf{x}}^i$ from Subsection A.11.1, but with respect to the subset $S_r$ and at scale $b_x$. Formally, for $\ell = r$ we have $x^i_{j_1,\ldots,j_{m'},\ell} = b_x$ if $j_t = 3$, and $j_k = 2$ for all $k \neq t$, and $x^i_{j_1,\ldots,j_{m'},\ell} = 0$ otherwise. Note that $\|\mathbf{x}^i\| = b_x$ for all $i \in [m]$.

Consider patches of dimensions $2 \times \ldots \times 2 \times L$ and stride 1. Thus, the set $\Phi$ corresponds to all the patches of dimensions $2 \times \ldots \times 2 \times L$ in the tensor. Note that since the last dimension is $L$, then the filter can "move" only in the first $m'$ dimensions. Also, note that there are $2^{m'} = n$ such patches. Indeed, given $(j_1,\ldots,j_{m'}) \in [2]^{m'}$, we can define a patch which contains the indices $\left\{(j_1,\ldots,j_{m'},0) + (\Delta_1,\ldots,\Delta_{m'},\Delta_{m'+1}) : (\Delta_1,\ldots,\Delta_{m'}) \in \{0,1\}^{m'}, \Delta_{m'+1} \in [L]\right\}$. We say that $(j_1,\ldots,j_{m'})$ is the *base index* of this patch. Note that each $(j_1,\ldots,j_{m'}) \in [2]^{m'}$ is a base index of exactly one patch. Also, if $(j_1,\ldots,j_{m'})$ includes some $r \in [m']$ with $j_r = 3$ then it does not induce a patch of the form $\left\{(j_1,\ldots,j_{m'},0) + (\Delta_1,\ldots,\Delta_{m'},\Delta_{m'+1}) : (\Delta_1,\ldots,\Delta_{m'}) \in \{0,1\}^{m'}, \Delta_{m'+1} \in [L]\right\}$, since for $\Delta_r = 1$ we get an invalid index.

We show that for any $\mathbf{y} \in \{0,1\}^m$ we can find a filter $\mathbf{w}$, such that $\mathbf{w}$ is an order-$(m'+1)$ tensor of dimensions $2 \times \ldots \times 2 \times L$ and satisfies the following. Let $N_{\mathbf{w}}$ be the neural network that consists of a convolutional layer with the patches $\Phi$ and the filter $\mathbf{w}$, followed by a max-pooling layer. Then, for all $i \in [m]$ we have: if $y_i = 0$ then $N_{\mathbf{w}}(\mathbf{x}^i) = 0$, and if $y_i = 1$ then $N_{\mathbf{w}}(\mathbf{x}^i) = 2\epsilon$. Thus, we can shatter $\mathbf{x}^1,\ldots,\mathbf{x}^m$ with margin $\epsilon$. Moreover, the spectral norm of the matrix $W$ that corresponds to the convolutional layer is at most $B$.

We now define the filter $\mathbf{w}$ of dimensions $2 \times \ldots \times 2 \times L$. For every $\ell \in [L]$ we define the components $w_{j_1,\ldots,j_{m'},\ell}$ as follows. Let $\mathbf{y}_{S_\ell} \in \{0,1\}^{m'}$ be the restriction of $\mathbf{y}$ to the indices in $S_\ell$. Then, $w_{j_1,\ldots,j_{m'},\ell} = \frac{2\epsilon}{b_x}$ if $(j_1,\ldots,j_{m'}) = \mathbf{1} + \mathbf{y}_{S_\ell}$, and $w_{j_1,\ldots,j_{m'},\ell} = 0$ otherwise. We show that for all $i \in [m]$, if $y_i = 0$ then $N_{\mathbf{w}}(\mathbf{x}^i) = 0$, and if $y_i = 1$ then $N_{\mathbf{w}}(\mathbf{x}^i) = 2\epsilon$. Suppose that $i \in S_r := \{k_1,\ldots,k_{m'}\}$ for some $r \in L$, and that $i = k_t$, i.e., $i$ is the $t$-th element in the

subset $S_r$. Then, the tensor $\hat{\mathbf{x}}^i$ has a non-zero component only at $x^i_{j_1,\ldots,j_{m'},r}$ with $j_t = 3$, and $j_s = 2$ for all $s \neq t$. Moreover, the filter $\mathbf{w}$ has a non-zero component at index $(q_1,\ldots,q_{m'},r)$ iff $(q_1,\ldots,q_{m'}) = \mathbf{1} + \mathbf{y}_{S_r}$. Hence, the inner product between $\mathbf{w}$ and a patch of $\mathbf{x}^i$ is non-zero iff the patch has a base index $(j_1,\ldots,j_{m'})$ such that $(j_1,\ldots,j_{m'}) + \mathbf{y}_{S_r} = (p_1,\ldots,p_{m'})$ where $p_t = 3$, and $p_s = 2$ for all $s \neq t$. If $y_i = 0$ then the $t$-th component of $\mathbf{y}_{S_r}$ is 0, and there is no patch such that the base index satisfies $j_t + (\mathbf{y}_{S_r})_t = j_t + 0 = p_t = 3$. Therefore, $N_{\mathbf{w}}(\mathbf{x}^i) = 0$. If $y_i = 1$ then the patch whose base index satisfies $j_t = 3 - (\mathbf{y}_{S_r})_t = 3 - 1 = 2$, and $j_s = 2 - (\mathbf{y}_{S_r})_s$ for $s \neq t$, gives output $\frac{2\epsilon}{b_x} \cdot b_x = 2\epsilon$ (and all other patches give output 0).

It remains to show that the spectral norm of the matrix $W$ that corresponds to the convolutional layer with the filter $\mathbf{w}$ is at most $B$. Thus, we show that for every input $\mathbf{x} \in \mathbb{R}^d$ with $\|\mathbf{x}\| = 1$ the inputs to the hidden layer are a vector with norm at most $B$. We view $\mathbf{x}$ as the vectorization of a tensor $\hat{\mathbf{x}}$ with components $x_{j_1,\ldots,j_{m'},\ell}$ for $(j_1,\ldots,j_{m'},\ell) \in [3]^{m'} \times [L]$. The inner product between a patch of $\mathbf{x}$ and the filter $\mathbf{w}$ can be written as

$$\sum_{\ell \in [L]} \frac{2\epsilon}{b_x} \cdot x_{q_1^{(\ell)},\ldots,q_{m'}^{(\ell)},\ell} \, .$$

Thus, for each $\ell$ there is a single index of $\hat{\mathbf{x}}$ that contributes to the inner product, since for every $\ell$ the filter $\mathbf{w}$ has a single non-zero component, which equals $\frac{2\epsilon}{b_x}$. By Cauchy–Schwarz, the above sum is at most

$$\frac{2\epsilon}{b_x} \cdot \sqrt{L} \cdot \sqrt{\sum_{\ell \in [L]} x^2_{q_1^{(\ell)},\ldots,q_{m'}^{(\ell)},\ell}} = \frac{2\epsilon}{b_x} \cdot \frac{b_x B}{2\epsilon} \cdot \sqrt{\sum_{\ell \in [L]} x^2_{q_1^{(\ell)},\ldots,q_{m'}^{(\ell)},\ell}} = B \cdot \sqrt{\sum_{\ell \in [L]} x^2_{q_1^{(\ell)},\ldots,q_{m'}^{(\ell)},\ell}} \, . \quad (18)$$

Hence, the input to the hidden neuron that corresponds to the patch is bounded by the above expression. Moreover, since for every $\ell \in [L]$ the filter $\mathbf{w}$ has a single non-zero component such that the last dimension of its index is $\ell$, then for every two patches with different base indices, the bound in the above expression includes different indices of $\hat{\mathbf{x}}$. Namely, if the inner product between one patch of $\mathbf{x}$ and the filter $\mathbf{w}$ is $\sum_{\ell \in [L]} \frac{2\epsilon}{b_x} \cdot x_{q_1^{(\ell)},\ldots,q_{m'}^{(\ell)},\ell}$ and the inner product between another patch of $\mathbf{x}$ and the filter $\mathbf{w}$ is $\sum_{\ell \in [L]} \frac{2\epsilon}{b_x} \cdot x_{p_1^{(\ell)},\ldots,p_{m'}^{(\ell)},\ell}$, then for every $\ell$ we have $(q_1^{(\ell)},\ldots,q_{m'}^{(\ell)}) \neq (p_1^{(\ell)},\ldots,p_{m'}^{(\ell)})$. Since by Eq. (18) the square of the input to each hidden neuron can be bounded by $B^2 \cdot \sum_{\ell \in [L]} x^2_{q_1^{(\ell)},\ldots,q_{m'}^{(\ell)},\ell}$ for some subset $\left\{ x_{q_1^{(\ell)},\ldots,q_{m'}^{(\ell)},\ell} \right\}_{\ell \in [L]}$ of components, and since for each two hidden neurons these subsets are disjoint, then the norm of the vector of inputs to the hidden neurons can be bounded by

$$\sqrt{B^2 \cdot \sum_{k \in [d]} x_k^2} \leq \sqrt{B^2 \cdot 1} = B \, .$$