# OpenReview forum: "The Sample Complexity of One-Hidden-Layer Neural Networks"
_NeurIPS.cc/2022/Conference — NeurIPS 2022 Accept_

### Official Review · Reviewer_6M6u · 2022-07-03

**Rating:** 6
**Confidence:** 3
**Soundness:** 3 good
**Presentation:** 3 good
**Contribution:** 3 good

**Summary:**

This paper studies the sample complexity of simple neural network model classes. The focus is on upper and lower bounds under various norm constraints on the network coefficients. It is shown that a spectral norm constraint is not sufficient to obtain bounds that are independent of the network width. To prove this, the authors estimate the fat shattering dimension of the considered hypothesis class. The surprising aspect of this result is that it does not hold if the activation is the identity but crucially depends on the nonlinearity. As a second contribution the authors show that a Frobenius norm constraint yields hypothesis classes whose sample complexity is independent of the network width. Finally, the authors identify two interesting settings where a spectral norm constraint is sufficient for getting bounds independent of the width.

**Questions:**

I would like to ask the authors to comment more on the novelty of their proof techniques.

**Limitations:**

This has been adequately addressed

**Strengths And Weaknesses:**

I think that this is an interesting paper, well written, and adding to the current body of knowledge on understanding the generalization behaviour of deep learning algorithms.

In terms of originality I am not sure if the paper introduces new proof techniques; maybe they can elaborate the novelty of their technical and mathematical contributions a bit more?

---

> ### Author Response · Authors · 2022-07-30
> **Response**
>
> Thanks for your comments!
> There are many Rademacher and fat-shattering bounds in the literature, but as far as we know they do not involve the smoothness of the activations in the way that we do it here.  For example, our lower bounds pinpoint the non-smoothness at the origin as a key factor in the "blowing up" of the norm of the relevant matrix, and theorems 4 and 5 involve new technical tricks to analyze polynomial activations, which may be of independent interest (e.g., lemmas 3 and 4). Finally, our lower bound for convolutional networks relies on a novel construction where each input is a vectorization of a tensor, which might be of independent interest.

---

### Official Review · Reviewer_UHEf · 2022-07-09

**Rating:** 7
**Confidence:** 3
**Soundness:** 3 good
**Presentation:** 2 fair
**Contribution:** 3 good

**Summary:**

Based on bounded norms of the weight matrices, the authors investigated the upper and lower bound sample complexity for neural networks with one hidden layer in this research.  They demonstrated that in contrast to bounding the Frobenius norm, however, bounding the spectral norm generally cannot result in size-independent guarantees - although it is not surprising that the spectral norm is insufficient to get width-independent sample complexity bound -  the paper and its theoretical analysis are very important.


The constructions did, however, highlight two situations in which the lower bounds can be avoided and a spectral norm control is sufficient to provide width-independent guarantees. The first scenario is when the activations are sufficiently smooth, and the second is in certain situations involving convolutional networks.

In general, this is a very important paper as deep neural networks still lack a very basic understanding of their behavior.

**Questions:**

Some definitions are missing. What is the definition of a Lipschitz function in your work? Is it the standard definition, (intuitively) a continuous function that is limited in how fast it can change?

Can you give more examples of other used Lipschitz functions?

Is σ(0) = 0, necessary in all the theorems, or it can be avoided (used for simplicity)?





**Limitations:**

some simple numerical experimentss would help justify the theory.

**Strengths And Weaknesses:**

Strengths:

1. A very solid theoretical paper - the theorems and proofs are exciting.
2. The paper deals with very important and required questions in the field of deep learning, not many papers focus on the theoretical side of deep learning - I find this paper a very important step towards better understanding DNNs.

Weaknesses:
Although it is clear that the paper is a theoretical paper, however, I have the following minor comment. The writing could still be improved, giving some intuitive explanation and details as to what and why each theorem holds and what each theorem mean -- it took some time to understand the theorems and the details.

---

> ### Author Response · Authors · 2022-07-30
> **Response**
>
> Thanks for your comments!  We will polish as suggested.
>
> -Lipschitz: Yes, it is the standard definition.  Essentially all commonly used activations are Lipschitz on bounded domains, and 1-Lipschitz after rescaling.
>
> - sigma(0)=0: Good question, and we indeed discuss it in the paper, see remark 1.

---

### Official Review · Reviewer_NAae · 2022-07-11

**Rating:** 6
**Confidence:** 3
**Soundness:** 3 good
**Presentation:** 4 excellent
**Contribution:** 2 fair

**Summary:**

This paper studies the norm-based uniform convergence bounds for two-layer neural networks. Their results give a tight understanding of spectral-norm-based uniform convergence. In particular, they proved that spectral norm is not sufficient for general settings to get a uniform convergence result. However, for NNs with certain smoothness conditions or some convolution structures, the spectral norm is sufficient.

**Questions:**

The uniform convergence guarantees derived in Sections 4 and 5 are size-independent. I wonder whether they also work for infinite width neural networks such as mean-field settings?

**Limitations:**

The authors have addressed their work's limitations and potential negative social impact.

**Strengths And Weaknesses:**

Overall, this paper is well-written and clear. The authors show that, in general, bounding the spectral norm cannot lead to size-independent guarantees. This negative result is quite interesting and insightful. However, I have a concern about the significance of the results. Since the size of the parameters is known through the training, it seems unnecessary to get a size-independent guarantee. Moreover, the spectral-norm-based convergence result seems hard to apply both empirically and theoretically, which may limit the application of the convergence result.

---

> ### Author Response · Authors · 2022-07-30
> **Response**
>
> Thanks for your comments!
>
> In our view, the main purpose of these type of size-independent bounds is to (1) help us understand how these networks can generalize regardless of how large they are; and (2) what kind of weight control (in terms of norms) affect their generalization.
>
> Regarding your question: Yes, the proof techniques are applicable to networks of unlimited size.

---

### Official Review · Reviewer_poyY · 2022-07-23

**Rating:** 7
**Confidence:** 5
**Soundness:** 4 excellent
**Presentation:** 4 excellent
**Contribution:** 3 good

**Summary:**

The paper presents various upper bounds (Rademacher complexity based) and lower bounds (fat shattering dimension based) on single hidden layer neural networks. These bounds shed light on the question whether the bounded spectral norm of the weight matrix is sufficient for having width independent uniform convergence guarantees. Moreover, there are similar discussions for Frobenius norms and input dimension dependence. The paper considers both generic and convolutional neural networks.

Contributions of the paper are summarized in 8 theorems:

* Theorem 1 shows that the fat shattering dimension is scaled with the network width for nonsmooth activation function, if only the spectral norm is bounded.
* Theorem 2 generalizes Golowich et al 2018 and Neyshabur et al 2015 to show that Frobenius norm bound is sufficient to bound the sample complexity for Lipschitz activation functions (width independent bound) using Rademachar complexity analysis.
* Theorem 3 shows that the fat shattering dimension for Frobenius norm bounded networks is input dimension dependent (for smaller input dimensions).
* Theorem 4 shows that the spectral norm is sufficient to bound the sample complexity for polynomial activation functions.
* Theorem 5 extends the result to the multilayer case for polynomial activation functions of type $z^k$. Lemma 4 and 5 are crucial for the proof of Theorem 4 and 5.
* Theorem 6 shows that the spectral norm is sufficient for convolutional networks with linear last layer.
* Theorem 7 shows that Rademacher complexity bound for convolutional networks with pooling  has logarithmic dependence on width.
* Theorem 8 shows, using fat shattering dimension, that the logarithmic width dependence is unavoidable.


**Questions:**

* Regarding Theorem 1: Anthony and Bartlett 1999 also provide an expression of fat shattering dimension although with slightly different assumptions (Theorem 14.18) and limitations. For example, the bound is input dimension dependent, the class of activation functions is different, etc. It would be good to add a discussion regarding this to the related works.
* Regarding Theorem 2: the proof, like Theorem 5 of Golowich et al, seems to be based on McDiarmid’s lemma and not relying on Dudley’s integral, which is also used to bound the Rademacher complexity using covering numbers. Is there any reason behind why Dudley’s integral is not used? Does it provide worse result if applied to this context with Lipschitz activation function? This can be an interesting theoretical insight and valuable to add.
* Theorem 4 and 5 prove that for polynomial activation functions, the spectral norm bound is sufficient for uniform convergence bound. However, the question remains if the lower bound, based on fat shattering dimension, can be further improved for these activation functions. A comment on this point can be a good add to the paper, if it is straightforward.
* The authors mention that they build on Ledent et al 2021 for Theorem 7. However, it is not currently clear, neither in the main paper nor in A.8,  what is the main difference between Ledent et al 2021. The difference in the respective proofs and bounds should be specified.
* If $[M]_{+}$ is the ReLU applied on $+/- 1/\sqrt{n}$, I cannot see why it “is just the constant matrix with value $1/\sqrt{n}$ at each entry” (page 5). It will be a combination of 0 and $1/\sqrt{n}$.
* I felt that Remark 4 is a bit confusing, although I got the point: namely, the presence of d in the lower bound does not contradict Theorem 2, since it assumes that “the above expression” is bigger than cd (according to the theorem) meaning that $(bBb_x/\epsilon)\sqrt{d}$ is bigger than $cd$, and therefore, d is smaller than $(bBb_x/\epsilon)^2$. I think this remark can be re-written. I suggest adding equation and theorem labels and refer to them explicitly instead of using demonstratives.
* The authors mention that “when the network employs weight-sharing”, it is possible to get width independent bound (actually logarithmic dependence for CNNs with pooling). This is not fully precise. The term $O_\Phi$ hides the dependence on kernel size. Consider a discrete circular convolution for which the matrix $W$ is a circulant matrix. In this case, $O_\Phi = n$. Even for a vanilla convolutional kernel, $O_\Phi$ is equal to the kernel size, which is a more intuitive notion of width for convolutional networks. This is clear at the end of A.7 right above A.8 where two summations range over m and n inside the square root.
* How do the current results change (or not) if one introduces a margin-based loss? For classification tasks, if one disregards the margin, it is possible to rescale the last layer without changing the classification accuracy, which would mean arbitrarily changing the product of spectral norm. Many previous works, e.g., Bartlett et al 2017, consider margin-based bounds. Why is this not considered in this work?


**Limitations:**

As mentioned by the authors themselves, there is a debate whether uniform convergence can explain generalization in neural networks mainly supported by some carefully designed experiments. It is not clear how one should understand the bounds of this paper in light of papers like Nagarajan et al 2019.

**Strengths And Weaknesses:**

**Strengths:**

From technical perspective, the paper provides a valuable contribution by clear exposition of the role of spectral and Frobenius norm in uniform convergence results.
The paper is well written. Given the sheer volume of works on explaining generalization error for neural networks, the authors do a good job connecting the results to the existing works for instance in Remark 2 on Bubeck et al 2021 or Remark 3 on implicit regularization. The paper contains many theoretical results of interest, combines many ideas from previous works and provide new proof techniques notably Theorem 4 and 5 (to the best of my knowledge). The proofs are well presented and are sound. Overall, this is a great paper and I enjoyed reading it.

**Weaknesses:**

* Although this might seem like a gratuitous comment and request, I think having numerical support, even a toy example, can be good to support the theory.
* I have some concerns about O_\Phi term in Theorem 4 (see below).
* I feel that the authors do not adequately comment about implications of the current result for a general theory of generalization for deep learning. Covering all existing works are of course not possible, however, works like Nagarajan et al 2019, or Zhang et al ICLR 2017 are widely discussed. It is at least expected that the authors clarify in which regime their bounds are applicable and do not suffer from examples of those papers.

---

> ### Author Response · Authors · 2022-07-30
> **Response**
>
> Thanks for your detailed comments!  We will make revisions accordingly. Below are some specific answers to your questions.
>
> - $O_\Phi$ term -- Note that for a circular convolution, this term equals the kernel size *divided by* the stride, so it can range anywhere from the kernel size to 1 (in the well-studied case of disjoint patches). In any case, it generally does not equal n in the notation used in our paper (i.e., the number of patches, unless some coordinate appears in all patches). We'd be happy to further clarify these points in the text as needed.
> - Theorem 2: it was simply more convenient for us to rely on theorem 5 from that paper. An entropy integral could probably be employed as well.
> - Theorem 4 and 5: Since our lower bounds apply specifically to nonsmooth activations, we're not sure in which sense it can be improved for polynomial ones.  We'd be happy to add a comment as suggested, if you can please elaborate.
> - Theorem 7: Ledent et al. (2021) considered a slightly different setting (which is more general, but with some technical differences). We use the same technique to obtain a simple proof for our setting. As we mentioned in the paper, the main novel contribution in Subsection 5.2 is Thm. 8.
> - $[M]_+$: Note that the example in the submission is with the absolute value activation, not ReLU. The assertion about $\sigma(M)$ being large is also true for ReLU, but the proof is less immediate.
> - Margin-based bounds: all our upper bounds (whose norm-based form is also common in the literature) can be easily converted to margin-based bounds by considering a composition of the hypothesis class with an appropriate loss (indeed, this is the approach taken for example in Bartlett et al 2017). In any case, we will add a discussion.

---

> > ### Comment · Reviewer_poyY · 2022-08-08
> > **Response to the authors**
> >
> > I would like to thank the authors for their comments and answers.
> >
> > * I still appreciate if a comment can be added regarding $O_{\Phi}$ to the main paper (or the supplementary material).
> > * Let me clarify my question regarding Theorem 4 and 5. The authors present lower bounds that holds for non-smooth activation functions and shows the spectral norm is in general insufficient. Theorem 4 and 5 show that the spectral norm can be sufficient to find an upper-bound the GE for polynomial activation functions. My question was: is there any lower bound for polynomial activation functions, basically, showing that spectral norm is also necessary for smooth activation functions?
> > * I also appreciate if the author could add more comments about the implication of their work for the discussions in  Nagarajan et al 2019. It would be valuable for the research community. For example, why should norm-based bounds be further studied in light of Nagarajan et al paper, or even the paper "Fantastic generalization errors and ..."?
> >
> > Otherwise, I had already voted for accepting the paper, and these suggestions are my preference to strengthen the paper.

---

> > > ### Author Response · Authors · 2022-08-08
> > > **Response**
> > >
> > > Thanks! We will make sure to incorporate these good suggestions.
> > >
> > > Regarding your question (spectral norm is also necessary for smooth activation functions?): The answer is yes. One can see this already for the identity activation function, in which case the hypothesis class studied in theorems 4+5 is equivalent to the class of linear predictors with Euclidean norm <= bB. By classical results on the sample complexity of linear predictors, a dependence on bB (and hence on the spectral norm bound B) is inevitable. We'd be happy to add a discussion of this point. Also, if we still misunderstood your question, please let us know and we'd be happy to explain further.

---

### Meta-Review · Area_Chair_bMWf · 2022-08-27

**Recommendation:** Accept
**Confidence:** Certain

**Metareview:**

The paper proves a novel, tighter bound norm-based bound for the generalization error of two-layer networks. All the reviewers agree that this is an important theoretical result and should be accepted.

**Award:**

No

---

### Decision · Program_Chairs · 2022-09-14

Accept